# Operando recombination kinetics in perovskite nanocrystal films revealed by in situ time-resolved photoluminescence

Dandan Cao, Ziyue Jiao, Jie Gao, Yi Wang ✉, Xi-Cheng Ai & Jian-Ping Zhang

Perovskite nanocrystals have shown great promise for optoelectronic applications. Understanding charge recombination under operational conditions, such as continuous-wave photoexcitation, is crucial for advancing device performance. Although time-resolved photoluminescence is widely used to study recombination kinetics, its reliance on ultrafast pulsed excitation fails to replicate the operational conditions. Here we develop an integrated spectroscopy platform that enables simultaneous acquisition of steady-state and time-resolved photoluminescence under continuous-wave illumination, with a wide range of photoexcitation intensity modulation. This approach resolves the long-standing mismatch between the typical time-resolved photoluminescence data and the actual continuous-wave operational behavior of perovskite nanocrystals. In contrast to the established Auger recombination model, which suggests accelerated recombination at high pump fluence, we demonstrate that the operando recombination kinetics is governed by the charge-carrier and trap-state interaction. This clarified recombination mechanism provides insight for designing perovskite nanocrystal films applied to efficient and stable light emission under high-power photoexcitation.

Metal halide perovskite nanocrystals (PNCs), also known as perovskite quantum dots, represent a novel class of nanomaterials that exhibit a combination of efficient photoluminescence (PL)[1], tunable bandgap[2], and strong light absorption[3]. These prominent characteristics, in conjunction with the simple preparation procedures,[4] render them highly promising for use in diverse optoelectronic devices[5,6]. In addition to the significant advances in material design and device fabrication, considerable efforts have been also made over the past decade to elucidate the unique physicochemical properties of PNCs, such as excellent defect tolerance[7] and drastic polaronic features[8], aiming to further extend the utilization of PNCs in specific practical applications.

Time-resolved PL (TRPL) techniques, including streak camera[9,10], fluorescence upconversion[11], and time-correlated single photon counting (TCSPC)[12] are hitherto the most widely used approaches for investigating the energy conversion mechanism of PNCs. By analyzing the dynamics or kinetics of photogenerated excitons and charge carriers following the excitation of ultrashort optical pulses, one can extract detailed information about the radiative and nonradiative charge recombination processes[13,14]. However, despite the dramatic advantages, such as high temporal resolution, tunable detection window, and superior sensitivity[15,16], it is recognized that typical TRPL techniques are perhaps incapable of revealing the landscape of charge recombination in PNC devices under real operational conditions. For example, color-conversion PNC devices are pumped by a stable continuous-wave (CW) light, while the TRPL measurements are performed under the excitation of ultrafast pulses. It is uncertain whether the observed phenomenon in TRPL experiments can accurately mirror the stable operational condition. At a comparable pump fluence, for instance, the peak power density of the ultrafast pulse is several orders of magnitude stronger than that of CW light. Therefore, a series of nonlinear optical processes can be detected by TRPL, while their occurrence is much less probable in real devices. This discrepancy

Key Laboratory of Advanced Light Conversion Materials and Biophotonics, School of Chemistry and Life Resources, Renmin University of China, Beijing, P.R. China. ✉ e-mail: ywang@ruc.edu.cn

poses a substantial challenge in certifying the strategy for improving material performance directly based on TRPL experimental results.

In the present study, we develop an integrated, in situ spectroscopy system that allows for the synchronous detection of steady-state PL and TRPL data. The sophisticated design of the photoexcitation method enables precise examination of the operando exciton/charge recombination kinetics of PNCs under the sustained excitation of a CW beam. As a proof of concept, the TRPL kinetics of photostable, close-packed PNC films were investigated by varying the CW excitation intensity over twentyfold. The standard steady-state PL spectra display notable discrepancies with the data of the typical TRPL measurement, whereas they are in excellent agreement with the in situ TRPL results. Specifically, with the enhancement of excitation intensity from 0.64 mW/cm² to 12.74 mW/cm², a superlinear increase in PL intensity accompanied by a prolongation of PL lifetime is recognized, which can be attributed to the charge-carrier and trap-state interactions, contradicting the widely established Auger recombination model. The operando charge recombination landscape, as elucidated by the proposed in situ TRPL technique, clarifies the efficacy of PNC films in the application of light-emitting devices working under high-power photoexcitation.

## Results

To begin this study, we first synthesized colloidal cesium lead bromide (CsPbBr₃) PNCs through the reported hot-injection approach with slight modifications[4]. The transmission electron microscopy (TEM) image reveals cubelike nanoparticles (Fig. 1a) with an average edge length of 10.4 nm (Supplementary Figs. S1 S2). An interplanar spacing of 0.42 nm is evidenced by high-resolution TEM (HR-TEM) (inset of Fig. 1a), in great consistency with the (110) plane of the cubic phase of CsPbBr₃. Figure 1b illustrates the steady-state UV-vis absorption and PL emission spectra of the as-synthesized PNCs dispersed in n-octane. The absorption band edge (~2.429 eV) is inferred from the minimum of the second derivative of absorbance, as depicted in the inset of Fig. 1b. The PL spectrum exhibits a single peak centered at ~516 nm (corresponding to 2.402 eV) with a small full-width-at-half-maximum of 19 nm. The Stokes shift, defined as the energy difference between the absorption band edge and the PL peak, is thus determined to be ~27 meV, in line with the direct bandgap structure of perovskite. The corresponding TRPL kinetics of the PNC solution measured by the typical TCSPC technique with varied pump fluence are presented in Fig. 1c. Prominent self-similarity is observed for TRPL decay profiles with a pump fluence below ~10 nJ/cm², as attributed to the single exciton recombination; by contrast, an additional rapid decay component gradually emerges at elevated pump fluence (see the inset),

which has been assigned to the Auger recombination of multiexciton[17]. Figure 1d shows the plot of PL intensity versus pump fluence, displaying an unambiguous derivation from the linear dependence at high pump fluence. This is in excellent agreement with the occurrence of non-radiative Auger recombination that reduces the photoluminescence quantum yield (PLQY) as mentioned above.

Now we move forward to close-packed PNC films that are applied for realistic optoelectronic devices. The films are fabricated by spin-coating the purified PNC colloidal solutions onto a quartz substrate, subsequently followed by thermal annealing and preillumination treatments, as schematically illustrated in Fig. 2a. Thermal annealing is beneficial for regulating the optical and electronic properties by intentionally sintering the nanoparticles, which has been well-established for both classical binary nanocrystals[18,19] and lead halide PNCs[20]. Scanning electron microscopy (SEM) images reveal that the particle size dramatically increases from ~10 nm (Fig. 2b) to ~100 nm (Fig. 2c) following thermal annealing, while the cross-section thickness (Supplementary Fig. S3) and PL homogeneity (Supplementary Fig. S4) almost remain unaffected. The intensified and narrowed peaks identified in the X-ray diffraction (XRD) patterns, as shown in Fig. 2d, are indicative of an increase in grain size and an improvement in crystallinity. However, it also brings about a decrease in PLQY (Supplementary Fig. S5), which is attributed to the loss of surface ligands during annealing that increases the density of trap states[21]. To overcome this issue, the annealed PNC film is exposed to the irradiation of a CW beam (wavelength: 450 nm, power density: 14 mW/cm²) for 30 minutes prior to further optical measurement. It is anticipated that such a pre-illumination treatment can reconstruct the surface/boundaries of perovskite grains, thus improving the PL efficiency and stability of the film[22–24]. The comparison of PL emission spectra summarized in Fig. 2e shows a remarkable PL redshift as a result of size expansion upon thermal annealing; on the other hand, the preillumination treatment does not alter the PL profile, while it gives rise to an obvious enhancement in the PL emission intensity (Supplementary Fig. S6). Figure 2f displays the PL intensity evolution versus illumination time for the three types of PNC films, as derived from the long-term in situ PL spectroscopy measurement (Supplementary Fig. S7), which unambiguously manifests the efficacy of the sequential annealing and pre-illumination treatment on optimizing the photostability.

The integrated in situ PL spectroscopy technique proposed for interrogating the operando charge recombination kinetics of PNC films is presented in Fig. 3a. In brief, a CW beam and a high-frequency pulse beam, which are respectively generated from the continuous and picosecond laser diodes, are directed to the same position of the tested sample. A pair of neutral density filters (NDFs) is applied to

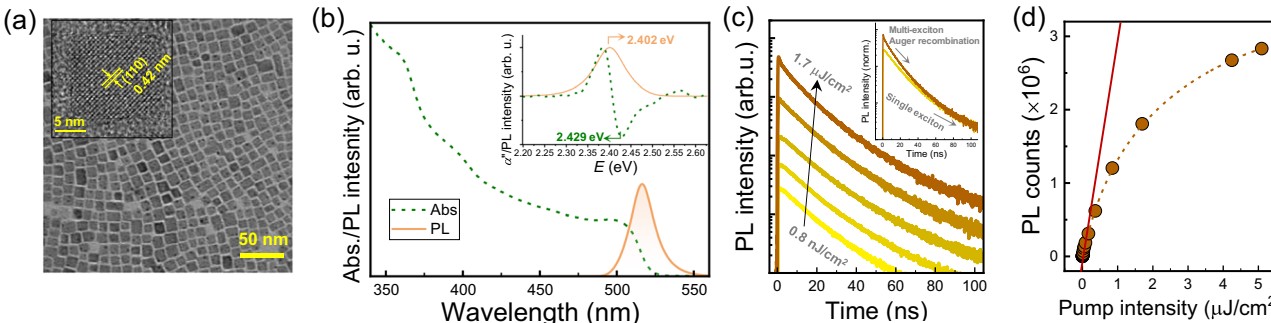

**Fig. 1 | Structural and optical characterization of CsPbBr₃ PNCs. a** TEM image of CsPbBr₃ PNCs. Inset: the corresponding HR-TEM image displaying an interplanar spacing of 0.42 nm. **b** Steady-state UV–vis absorption and PL emission spectra of CsPbBr₃ PNCs dispersed in n-octane. The inset displays the excitation energy-dependent second derivative of absorbance (α″) and PL intensity, resulting in a Stokes shift of 27 meV. **c** Typical TRPL profiles as a function of the pump fluence of

femtosecond pulses (from bottom to top: 0.8 nJ/cm², 3.4 nJ/cm², 51 nJ/cm², 0.17 μJ/cm², and 1.7 μJ/cm²). Inset: comparison of the normalized TRPL kinetics under the pump fluence of 0.8 nJ/cm² and 1.7 μJ/cm² respectively. **d** Dependence of steady-state PL intensity on the pump fluence of femtosecond pulses, where the red line shows the linear dependence at low pump fluence.

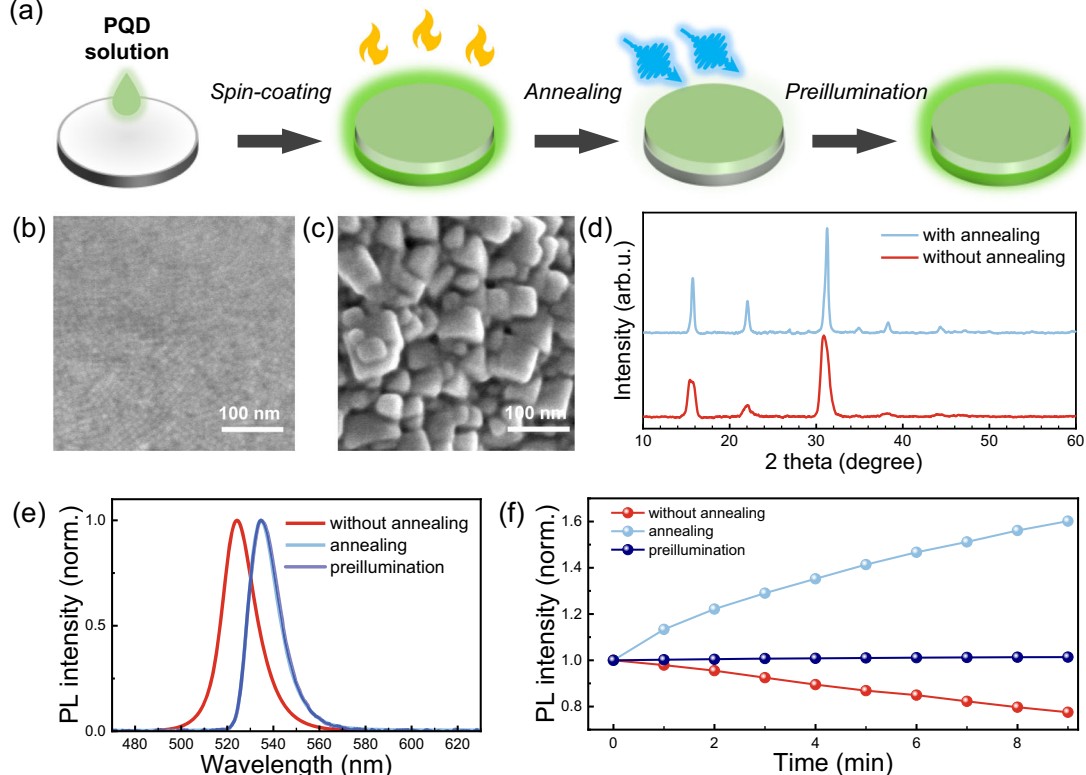

**Fig. 2 | Fabrication and characterization of photostable PNC films. a** Schematic representation of photostable PNC film fabrication that incorporates sequential spin-coating, thermal annealing, and preillumination processes. SEM images of the PNC films (**b**) before and (**c**) after the thermal annealing treatments and the corresponding (**d**) XRD patterns. **e** PL emission spectra and (**f**) PL stability for PNC films with different treatment procedures.

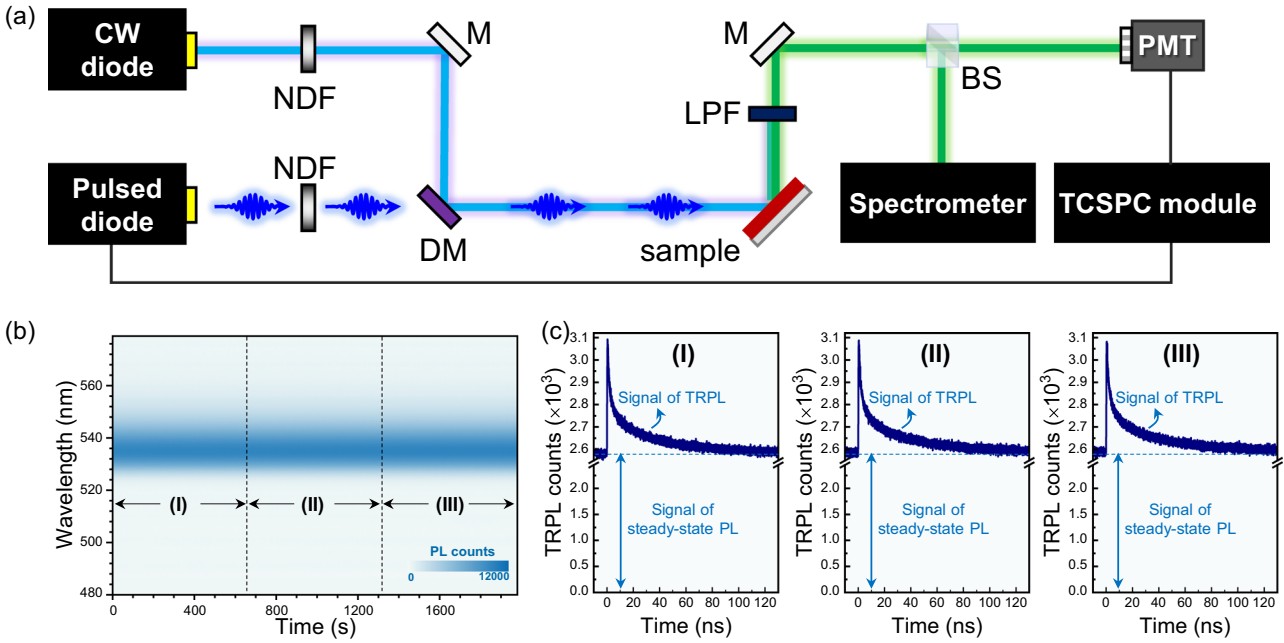

**Fig. 3 | In-situ PL characterization platform and representative results.**
**a** Schematic illustration of the integrated in in situ PL spectroscopy platform for the simultaneous acquisition of steady-state PL spectra and TRPL kinetics (see text for details). *BS* beam splitter, *DM* dichromatic mirror, *LPF* long-pass filter, *M* mirror, *NDF* neutral density filter, *PMT* photomultiplier tube. Representative experimental results of (**b**) in situ PL spectra and (**c**) TRPL kinetics of the PNC film measured in a synchronous manner.

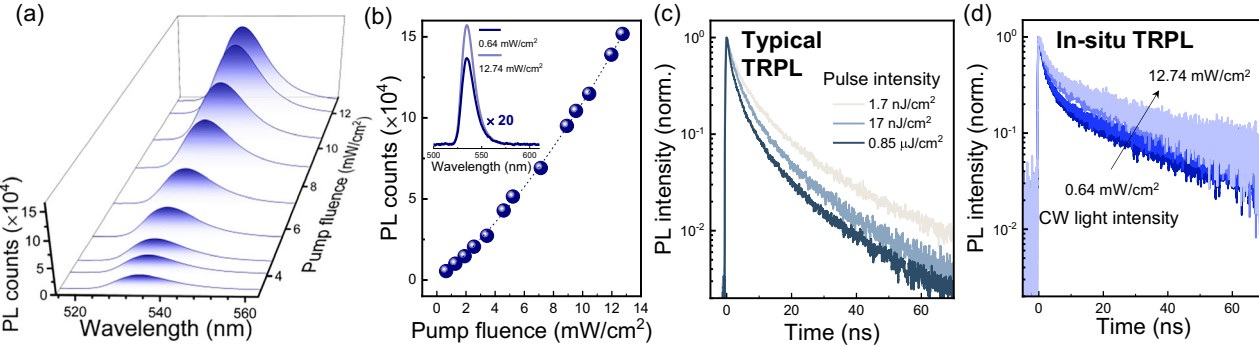

**Fig. 4 | CW pump fluence-dependent PL properties of the PNC film. a** Steady-state PL spectra of the PNC film measured by in situ PL spectroscopy at varied CW pump fluence. **b** Corresponding PL intensity versus CW pump fluence, which displays superlinear dependence. Inset: comparison of the PL spectra measured with excitation intensity values of 0.64 and 12.74 mW/cm² , respectively, where the former spectrum has been multiplied by a factor of twenty. Pump fluence-dependent PL decay kinetics of the PNC film measured by (**c**) typical TRPL and (**d**) in situ TRPL. The numbers denoted in panel (**c**) correspond to the fluence of the femtosecond pulse beam. In panel (**d**), the pump intensities of the CW beam for the presented TRPL data are 0.64, 1.27, 6.37, and 12.74 mW/cm², respectively. For the sake of comparison, the baselines of in situ TRPL data have been subtracted, and the peak intensities have been normalized.

accurately modulate the excitation intensity of the CW and pulse beams. By eliminating the excitation light with a long-pass filter (LPF), the pure PL signals are subsequently separated by a beam splitter (BS) and collected by a spectrometer and a photomultiplier tube (PMT) spontaneously. The spectrometer records steady-state PL spectra as is, while the PMT connected to a TCSPC module is employed to output TRPL responses. It is worth noting that the femtosecond laser, as employed in typical TRPL measurements, has been replaced with a picosecond laser diode used as the pulse beam source for in situ TRPL testing. The picosecond laser diode possesses the capability to operate at a much higher frequency (5 MHz) than the femtosecond laser (6 kHz), a crucial factor in ensuring the acquisition of high-quality in situ TRPL data without significantly prolonging the accumulation time. Figure 3b displays the as-obtained in situ PL spectra, which are centered at 535 nm regardless of illumination period, aligning closely with the standard PL spectrum shown in Fig. 2e. The fluctuation of PL intensity is found to be less than 2% over the entire experimental period (Supplementary Fig. S8), validating the prominent stability of the integrated PL spectroscopy platform. Concomitant with the in situ PL spectra measurement, TRPL results are obtained by means of TCSPC. Three sets of TRPL kinetics with the same accumulation time of ~ 660 s, corresponding to the three regions denoted in in situ PL spectra as exemplified in Fig. 3c, share the identical time-resolved features. Firstly, the TRPL traces demonstrate a substantial baseline (ca. 2600 counts in our case) arising from the steady-state PL generated by the CW beam, which signifies the capability of examining charge recombination kinetics under continuous photoexcitation. Secondly, the three sets of TRPL kinetics are self-similar with a variation of average lifetime as small as ~ 4% (Supplementary Fig. S9 and Supplementary Table S1), indicative of excellent reproducibility for the in situ TRPL test. More importantly, the picosecond pulse intensity is sufficiently weak, so as not to pose interference to the steady-state PL spectra measurement (Supplementary Fig. S10) nor to introduce any pump fluence-dependent TRPL behavior (Supplementary Fig. S11). This provides a guarantee that the in situ TRPL functions in a perturbation manner, whereby the operando condition is solely determined by the CW light intensity and not interfered by the pulse beam.

Having experimentally verified the reliability of the integrated in situ PL spectroscopy platform, we conduct a systematic exploration into the operando charge recombination kinetics of PNC films. A series of representative steady-state PL spectra with varied CW pump fluences are presented in Fig. 4a (see Supplementary Fig. S12 for complete data). The PL intensity increases with the pump fluence while the spectral profile remains consistent. A closer inspection uncovers a

superlinear relationship between the PL intensity and the CW pump fluence, as illustrated in Fig. 4b and its inset. For instance, with a multiplication of the PL signal measured at 0.64 mW/cm² by twenty, it is still significantly lower than that measured at 12.74 mW/cm². Such a superlinear dependence indicates that the PLQY is enhanced with the elevation of the excitation intensity. Interestingly, the typical TRPL data summarized in Fig. 4c elucidates accelerated decay kinetics at high pump fluence, which implies a more severe nonradiative exciton/charge recombination process, as is the case in PNC colloidal solution (Fig. 1c). However, this is at odds with the increase in PLQY at high pump fluence as derived from steady-state PL spectra, which helps exclude the multi-exciton Auger recombination. Conversely, as visible in Fig. 4d, the in situ TRPL data demonstrate a more gradual decay profile when the intensity of the CW light is increased from 0.64 mW/cm² to 12.74 mW/cm². Notwithstanding the underlying mechanism that will be discussed later, in situ TRPL kinetics are phenomenologically in accordance with the superlinear pump fluence-dependent PL spectra. The stark difference between the typical TRPL and in situ TRPL results signifies the necessity of performing experiments in a perturbation manner for precisely elucidating the operando mechanism, where the condition of the CW excitation is strictly kept constant for both steady-state and time-resolved measurements. Besides, the PNC film demonstrates highly reproducible spectroscopic and kinetic features following the complete in situ TRPL testing period (Supplementary Figs. S13, S14), thus validating that the observed CW intensity-dependent behaviors arise from the intrinsic properties of PNCs rather than the irreversible alterations posed by light degradation or chemical decomposition.

In order to gain deeper insights into the photophysics of operando charge recombination in PNC films, we first compare their in situ TRPL results to those measured in colloidal PNC solutions. As illustrated in Fig. 5a, the in situ TRPL kinetics of the PNC solution present negligible alteration in response to a wide range of CW pump intensity variations. No additional rapid decay component is observed at high pump fluence, in contrast to the typical TRPL data shown in Fig. 1c, indicating that the multiexciton effect is minimized under CW photoexcitation. The distinctive in situ TRPL behaviors facilitate the ascription of the retarded TRPL decay to the specific microscopic structure of PNC films, whose formation is accompanied by the transition from highly dispersed-state nanoparticles in solution (Fig. 1a) to sintered close-packed large grains in solid-state configuration (Fig. 2c). In the meantime, the improved PLQY at high CW pump fluence in PNC films is analogous to the findings reported in hundred-nanometer-sized polycrystalline perovskites[22].

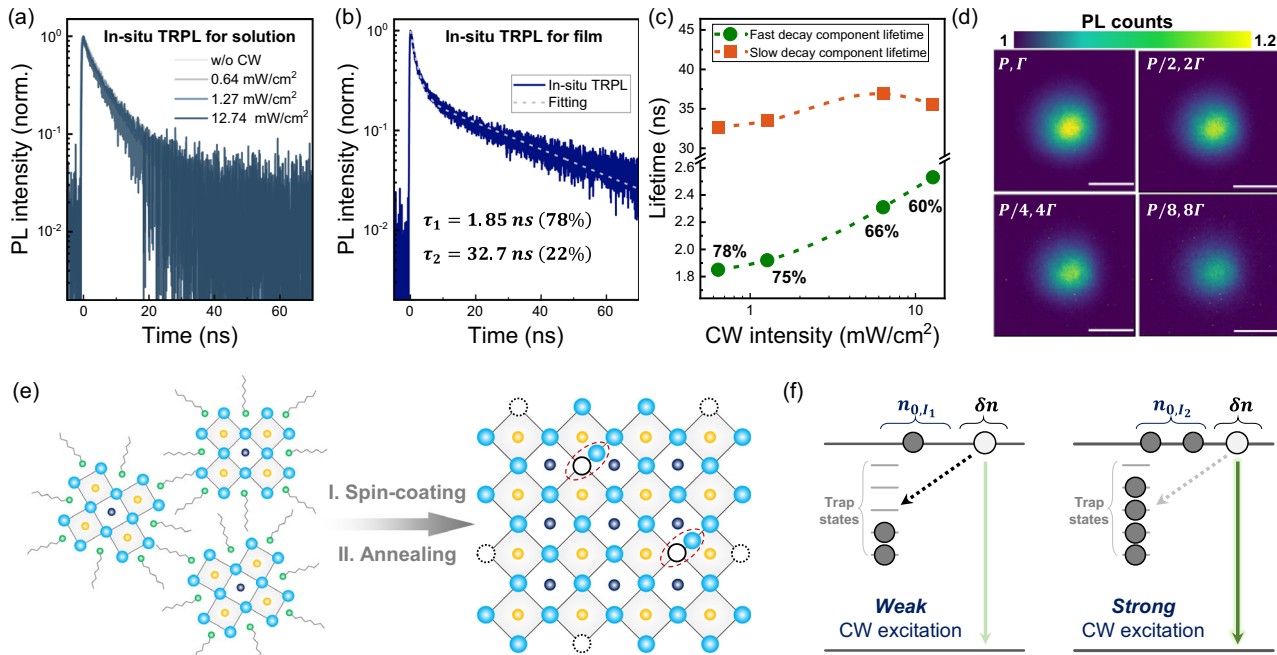

**Fig. 5 | Operando recombination mechanism derived from in situ TRPL and PL imaging analysis. a** CW pump intensity-dependent in situ TRPL kinetics for the colloidal PNC solution. **b** Representative in situ TRPL kinetics (solid line) of the PNC film measured at a CW pump intensity of 0.64 mW/cm$^2$ and the corresponding fitting results (dashed line), where $\tau_1$ and $\tau_2$ are the lifetimes of the fast and slow components derived from the biexponential function. **c** Lifetime values of the fast (green) and slow (orange) decay components as a function of CW pump intensity. Also denoted in percentage are the weights of the slow component for each set of in situ TRPL data. **d** Wide-field PL images of the PNC film measured with the desired excitation power and integrating time. Scale bar: 5 μm. Schematics for illustration of (**e**) the morphological evolution during PNC film fabrication and (**f**) the operando recombination mechanism with different CW excitation levels.

In quantitative, provided that multi-body Auger recombination is negligible, the equilibrium-state classical continuity equation in polycrystalline perovskites is typically expressed as: $dn_0/dt = G - k_1 n_0 - k_2 n_0^2 = 0$, where $n_0$ is the carrier density at the equilibrium state, $t$ is time, $G$ is the rate of photocarrier generation, $k_1$ is the rate constant for first-order nonradiative recombination, and $k_2$ is the rate constant for bimolecular recombination. Once a perturbation is superposed on the equilibrium state, the expression should be rewritten as: $d(n_0 + \delta n)/dt = G - k_1(n_0 + \delta n) - k_2(n_0 + \delta n)^2$, in which $\delta n$ is the density of excess carrier with a value much smaller than the equilibrium-state carrier density in a perturbation case ($\delta n \ll n_0$). A combination of the two aforementioned equations results in the classical continuity equation being reduced to the following form: $d\delta n/dt = -k_1 \delta n - 2 k_2 n_0 \delta n$, which is indicative of first-order reaction kinetics. In situ TRPL experiment fulfills the prerequisites of perturbation, therefore the time-dependent PL intensity is predicted to be described by the first-order kinetics model with a decay lifetime of $\tau$, $I_{PL}(t) = I_{PL}(0) \cdot \exp(-t/\tau)$. Furthermore, in consideration of the inhomogeneity of the chemical environment for individual perovskite grains, it has been proposed that the TRPL data be fitted more accurately with a bi-exponential function[25] or a stretched exponential function[26]. Here, we adopt the bi-exponential model for elucidation due to its higher fitting goodness than the stretched exponential model (Supplementary Fig. S15).

A representative in situ TRPL data in conjunction with the biexponential fitting results are depicted in Fig. 5b, where the fast ($\tau_1$) and slow ($\tau_2$) decay components have been suggested to arise from charge recombination at the grain interface/surface and within the volume of grains, respectively[27,28]. As the CW pump intensity is elevated from 0.64 to 12.74 mW/cm$^2$ (Fig. 5c), the value of $\tau_1$ increases by a factor of

37% (1.85 ns → 2.53 ns) while the variation in $\tau_2$ is less than 13% (see complete fitting results in Supplementary Fig. S16 and Supplementary Table S2). Concurrently, the weight of interface/surface recombination is reduced from 78% to 60%. It has been proposed that trap states are enriched at the grain interface/surface, where charge recombination occurs predominantly via a nonradiative path[27,28]. Therefore, the prolonged lifetime and decreased weight of the fast decay component of in situ TRPL kinetics at elevated CW pump fluence collectively demonstrate that trap state-mediated nonradiative charge recombination is suppressed by amplifying the intensity of photoexcitation. This is in excellent agreement with the observed superlinear behavior of CW intensity-dependent in situ PL spectra. By visualizing the light emission patterns, wide-field PL microscopy images (Fig. 5d) further verify the improved PL efficiency of PNC films upon a strong CW illumination. Specifically, the PL imaging experiments are conducted by simultaneously reducing the excitation power ($P$) and prolonging the integrating time ($\Gamma$) with the same factor. While the product of $P$ and $\Gamma$ remains constant, the PL intensity exhibits a prominent decrease with the reduction in $P$ (see detailed analysis in Supplementary Fig. S17), indicative of a higher PLQY by enhancing the CW illumination.

By taking all the aforementioned experimental results into account, the mechanism of charge recombination in PNC films under operational conditions can be excellently interpreted as below. As verified by TEM and SEM characterizations, PNCs undergo nanoparticle sintering to form large-size grains when the colloidal solution is subjected to spin-coating and thermally annealing to fabricate photostable, close-packed films (Fig. 5e). The sintered grains possess a size highly exceeding the exciton diameter of PNC (~ 7 nm)[4], endowing PNC films with new photophysical properties analogous to those of three-dimensional polycrystalline perovskite films[29]. It should be also noted that a substantial proportion of surface ligands is inevitably eliminated during the process of thermal annealing[21], which results in

the formation of additional trap states for perovskite grains[21,30]. These trap states can be distributed either at the grain interface/surface or within the volume[31], with the former being particularly significant in nonradiative charge recombination, due to the well-documented phenomenon of defect tolerance in perovskites[32]. The landscape of operando charge recombination revealed by in situ TRPL is illustrated in Fig. 5f, where $n_{0,I_i}$ and $\delta n$ denote the charge carriers generated by the CW beam and the pulse beam, satisfying the perturbation requirement, $\delta n \ll n_{0,I_i}$. The decay of $\delta n$ involves two pathways: trap state-mediated nonradiative charge recombination (dashed arrow) and radiative charge recombination (solid arrow), which correspond to the first and second terms, respectively, in the continuity equation $d\delta n/dt = -k_1 \delta n - 2k_2 n_0 \delta n$. With the elevation of the CW pump intensity, the value of $n_0$ is increased, which is anticipated to accelerate the TRPL decay; by contrast, the density of empty trap states (i.e., nonradiative recombination centers) is decreased owing to the trap filling by charge carriers, thus resulting in a reduced $k_1$ and a retarded recombination kinetics. Under operational conditions, the density of charge carriers is several orders of magnitude smaller than the trap state density[33–36]. This indicates that the effect of photoexcitation on filling trap states is dramatically greater than that on increasing $n_0$. As a consequence, the in situ spectroscopy results demonstrate a concomitant increase in the steady-state PL intensity and prolongation in the TRPL lifetime with the enhanced CW pump fluence, which mirrors the landscape of charge recombination in PNC films under realistic operational conditions that is not revealed by typical TRPL techniques.

In conclusion, the challenge of characterizing charge recombination in PNC films under continuous operational conditions is conquered through the proposal of an integrated in situ PL spectroscopy technique, which enables the acquisition of TRPL kinetics in a perturbation manner in conjunction with steady-state PL spectra measurements. The PL spectra manifest a superlinear dependence on the CW pump intensity, ranging from 0.64 mW/cm² to 12.74 mW/cm². Such a specific characteristic in close-packed films, distinct from that in colloidal solutions, contradicts typical TRPL results interpreted within the widely established framework of the Auger recombination model. Conversely, it aligns closely with the in situ TRPL data, which demonstrate a prolonged charge recombination by enhancing the CW pump fluence. With a combined analysis of the morphological and spectroscopic evolution in PNCs, we elucidate the fact that steady-state PL and TRPL behaviors can be collectively attributed to trap filling by charge carriers. The trap-filling effect suppresses nonradiative recombination with an increase in charge carrier density, according to the classical continuity equation, which retards the TRPL decay and improves the PL efficiency. The clarified operando charge recombination mechanism in PNC films highlights the significance of the interaction between photogenerated charge carriers and trap states on charge recombination kinetics, while the impact of Auger recombination under operation conditions is negligible, which justifies the efficacy of PNC films in light-emitting applications under high-power excitation levels.

## Methods
### Materials
Cesium carbonate (Cs₂CO₃, 99.99%, 3Achemical), Oleic acid (OA, AR, Mreda), Oleylamine (OAm, 90%, Rhawn), 1-octadecene (ODE, 90%, Heowns), Lead (II) bromide (PbBr₂, 99.999%, Sigma-Aldrich), Methyl acetate (MeOAc, 99%, Aignep). All chemicals were used without further purification.

### Preparation of Cs-oleate
Cs-oleate was prepared by combining Cs₂CO₃ (0.777 g), OA (4 mL), and ODE (36 mL) in a 100 mL three-neck round-bottom flask. The mixture was first stirred under vacuum at 120 °C for 30 minutes to eliminate moisture and oxygen, and then stirred under a nitrogen atmosphere until the complete dissolution of Cs₂CO₃.

### Synthesis of CsPbBr₃ PNCs
136 mg of PbBr₂, 10 mL of ODE, 2 mL of OAm and 2 mL of OA were loaded into a 50 mL three-neck round-bottom flask and subjected to vacuum drying at 80 °C for 1 h to remove residual moisture and oxygen. The reaction mixture was then heated to 175 °C under a continuous flow of nitrogen. Upon reaching the desired temperature, 0.8 mL of pre-prepared Cs-oleate solution that had been preheated at 100 °C was swiftly injected to initiate nucleation and growth of CsPbBr₃ nanocrystals. The reaction proceeded for 50 s before being rapidly quenched by the immersion in an ice-water bath to halt crystal growth. The as-synthesized PNC colloidal solution was then mixed with MeOAc at a volume ratio of 1:4 and centrifuged at 10,464 × g for 5 min. The resulting precipitate was redispersed in n-hexane and mixed again with MeOAc at a volume ratio of 2:1, which is followed by a second-round centrifugation at 10,464 × g for 5 min. The purified PNCs were redispersed in n-octane for optical characterizations and used as the precursor solution to fabricate PNC films.

### Fabrication of PNCs films
Quartz substrates were cleaned in an ultrasonic bath for 30 min with detergent, deionized water, acetone, and isopropanol, respectively, and then treated with ultraviolet-ozone (UV-ozone) for 15 min. Subsequently, 25 μL of PNC solution were spin-coated onto the quartz substrate at 2000 rpm for 30 s. For thermal annealing treatment, the PNC films were heated at 150 °C for 30 min in air. For preillumination treatment, the annealed films were irradiated with continuous-wave light (wavelength: 450 nm, power density: 14 mW/cm²) for 30 min.

### Structural characterization of PNCs
Transmission electron microscopy images were obtained on a FEI Talos F200X microscope with a 200 kV accelerating voltage. Dynamic light scattering measurement was conducted on a Malvern Zetasizer Nano ZS90 particle size analyzer. The surface morphology of the PNC films was characterized using a ZEISS GeminiSEM 300 field-emission scanning electron microscope, and the film thickness was determined by a Hitachi SU8010 field-emission SEM. X-ray diffraction patterns were obtained by using a Shimadzu XRD-7000 diffractometer with Cu Kα radiation performed in the 2θ range of 10° – 60° at a scan rate of 2° min⁻¹.

### Typical optical and spectroscopic characterizations
Ultraviolet-visible (UV–vis) absorption spectra were measured with an Agilent Cary 60 UV-Vis spectrophotometer operating at a scanning rate of 10 nm/s and covering a wavelength range of 300–800 nm. Steady-state photoluminescence (PL) spectra were measured by using an Edinburgh FLS 980 spectrometer, and the PL quantum yield was determined with the same instrument equipped with an integrating sphere. For typical time-resolved PL measurement, a femtosecond pulsed laser (PHAROS, 190 fs, 6 kHz), in conjunction with an optical parametric amplifier (OPA, ORPHEUS) were utilized to generate the excitation beam at 450 nm. The beam was attenuated by a neutral density filter to control its intensity and then directed to the sample. The as-generated TRPL signals were collected by a photomultiplier tube (Hamamatsu H7421) and recorded by a time-correlated single-photon counting (TCSPC) module (PicoQuant, PicoHarp 300). To achieve the wide-field PL microscopy image, the 457 nm-CW light by a laser diode (Changchun New Industries, MBL-U-457 nm) was focused on the surface of PNC films by the objective lens (Nikon, CF Plan Apo, 150X), whereupon the PL signals were collected by a CMOS camera (Intelligent Scientific Systems, IsCMOS-TRC411) for visualization.

### Integrated in situ PL spectroscopy measurement
The principle of the integrated in situ spectroscopy for the synchronous measurement of steady-state PL spectra and TRPL kinetics is illustrated in Fig. 3a. In detail, a continuous laser diode (Laserland,

Wuhan RLTC Technology Co., Ltd., 1875A-450D-80mw-5v) and a picosecond laser diode (Edinburgh Instruments EPL-405, ~100 ps, 5 MHz) were used to generate the CW beam (wavelength: 450 nm) and pulse beam (wavelength: 405 nm), respectively, whose intensities were strictly modulated by the neutral density filters (Daheng Optics, GCO-074M) to meet the criteria of perturbation. By using a dichromatic mirror (Lbtek DM10-425SP), the CW and pulse beams were directed to the same position of the tested PNC film. A long-pass filter (Daheng Optics GCC-300112) was adopted to remove the excitation signals, and the PL signals were separated by a beam splitter (Heng Yang Guang Xue HCBS1-025-70-VIS, splitting ratio: 7:3), whereby 70% of the signals were collected by a spectrometer (AvaSpec-ULS2048XL) and the remaining 30% of signals were recorded by a photomultiplier tube (Hamamatsu H7421) connected with the time-correlated single photo counting module (PicoQuant, PicoHarp 300).

## Data availability

The minimum per-figure dataset is provided in the accompanying Source Data file. The full raw datasets are available in the Figshare database (https://doi.org/10.6084/m9.figshare.30399451). Source data are provided in this paper.

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

## Acknowledgements

We gratefully acknowledge funding from the National Natural Science Foundation of China (NSFC) (Grant Nos. 22373114 and 22273119), Beijing Natural Science Foundation (Grant No. 2232008), and the Research Funds of Renmin University of China (22XNKJ08).

## Author contributions

Y.W. conceived the idea and supervised the project; D.C. dominated all the experiments; Z.J. provided helps in optical microscopy images; J.G. provided helps in steady-state spectroscopy measurements; D.C. and Y.W. drafted the manuscript; X.-C.A. and J.-P.Z. provided suggestions of the manuscript revision; Y.W. finalized the manuscript; all authors discussed the results and provided comments on the manuscript.

## Competing interests

The authors declare no competing interests.
