## [Peer review file · Nature Communications]

Operando Recombination Kinetics in Perovskite Nanocrystal Films Revealed by In-Situ Time-Resolved Photoluminescence

Corresponding Author: Professor Yi Wang

Version 0:

Reviewer comments:

Reviewer #1

(Remarks to the Author)

The paper by Cao et al. presents a spectroscopic study of quantum dot perovskite materials, with a focus on their potential use in color conversion layers. The authors perform time-correlated single-photon counting (TCSPC) measurements under steady-state illumination bias and conclude that multiple processes are involved in controlling the emission dynamics, including Auger recombination and photodegradation-related photochemistry.

The study gives me mixed feelings. On one hand, the general approach is interesting. Performing measurements under steady-state illumination bias is a promising way to understand the dynamics of materials under conditions relevant to actual device operation. On the other hand, instead of carefully presenting and analyzing their results, the authors often push forward broad, principle-challenging claims and present their data in a way that can be misleading. I find that the paper, in its current form, is not methodologically solid and would require substantial revision before it could be considered for publication. Furthermore, I feel the work does not meet the high standards expected for Nature Communications, and would be more suitable for a specialized journal such as Chemistry of Materials or Communications Chemistry.

My specific comments are as follows:

1. The authors describe their measurements as "operando dynamics." I believe this is misleading. The term would be appropriate if the processes they observed were reversible, and if steady-state illumination accurately simulated device operating conditions. However, many of the changes observed are irreversible, involving photodegradation and permanent chemical alterations. Therefore, the premise that the study represents true "operando" conditions is flawed. A major part of the introduction and context needs to be rethought and rewritten to reflect that the focus should be on photodegradation, not just dynamic processes during operation.
2. I am very disappointed by the presentation of Figures 2c, d, g, and h. It appears that the figures were designed in a way that could mislead an uncaredful reader. In comparing conventional TRPL and photoluminescence under illumination bias, the data should be presented on identical scales, and the excitation intensities should be normalized considering the excitation density in the sample. The authors fail to do this, instead presenting the datasets at very different scales and intensities to suggest that the results with and without bias are dramatically different — which is not true. After extracting and replottting the data myself, I find that the kinetics are actually very similar; only the trends with increasing illumination intensity differ somewhat. This is a serious flaw in data presentation.
3. Figure 4 clearly shows the effects of photodegradation and related permanent photochemical changes. However, the authors do not attempt to quantify the contribution of these irreversible changes compared to other dynamic effects such as multiparticle interactions and Auger processes. Without a quantitative analysis, it is very difficult to judge how significant the reported findings are. A thorough decomposition and estimation of different contributions are essential.
4. Similarly, Figure 5 is highly qualitative and does not allow the reader to assess the confidence level of the conclusions or sufficiently support the interpretation of the spectroscopic data. Figure 5 is just a cartoon, not adding much to the study or supporting it. A more rigorous, controlled investigation correlating structural changes (upon annealing or in some other way) to optical properties, using systematical and quantitatively well-defined sample sets, would be needed to substantiate the proposed interpretations. As it stands, the data do not have the level of rigor expected for a Nature Communications

publication.

Reviewer #2

(Remarks to the Author)

This manuscript reports on the spectroscopy of lead-halide perovskite quantum dots (QD) under operational conditions for LEDs. This is a very interesting and valuable manuscript and probably should be published. But there are many questions to address.

1. The authors use PL dynamics initially and then switch to kinetics. The initial point is that consistency in terms should be used. Pick one or the other. But it turns out that that the semantics matter. PL rarely measures dynamics. It mostly measures kinetics. There is a recent discussion of the differences between kinetics and dynamics¹. And there are recent examples of t-PL dynamics in LHP QD².
2. The introduction begins with the good features of LHP QD for LED. But why not CdSe and other covalent QD that were investigated first³? Why are LHP QD special. One answer is their unique lattice that is defect tolerant and does so by supporting liquid-like structural dynamics⁴⁻¹⁰.
3. The authors refer to t-PL measurements with streak cameras. They should cite the most recent and most detailed t-PL measurements of LHP QD that reveal their dynamics and kinetics for particles in dispersion², 11-17.
4. My main concern is that the annealed QD film is so different from the pristine film that it may not be meaningful to compare. The pristine film contains QD of 10 nm diameter. These are true QD. But the annealed film contains nanocrystals of 100 nm diameter which are not QD. Some NC are QD and some QD are NC but they are not the same as recently reviewed¹⁸. The main issue is the diameter relative to the exciton Bohr length which confers QD effects to excitonic structure and dynamics.
5. Fig1g shows the Abs spectra of the two films which look totally different. Why is there a more prominent excitonic peak for the annealed QD that increase in size to NC that are not QD.
6. Typo errors here: "Now we move forward to the PL mechanism of PQD films under the operational conditions of CCLs. Figure 2a displays 90 the steady-state PL emission spectra of the annealed film excited by a CW beam". It should be unannealed film
7. Fig2b. Why does the pristine film show a sublinear saturation curve for intensity? It should slow slightly due to XX formation. It may also slow due to hot exciton trapping at the surface which causes non-radiative recombination. The authors should do a PLE experiment and measure the QY as a function of energy to verify if there are hot exciton surface effects¹⁹, 20. Fig2c suggests that there is faster nonradiative decay by an excitation induced process.
8. Fig2. The in situ t=PL is different from the dispersion. Why is this? Mainly the in situ is single exponential whereas the dispersion is bi-exponential as is commonly seen. Why the single exponential?
9. Fig2c and d should be plotted on the same time window for better comparison
10. Fig2e shows a shape change to the spectrum. It looks like there is an ASE peak. Is there a peak developing? Perhaps a subtraction procedure would be useful.
11. I am confused about the description of data in Fig2. The ex-situ data are in a film or a dispersion? Is it that in site and ex situ are for a film or for a device?
12. The t-PL kinetics should be fit to a biexponential and report of the two time constant and amplitudes.
13. They switch from a fs pulse to a ps pulse without much explanation. Why was this done? Where are the pulse durations and pump wavelengths? Both should be reported. They discuss the non-perturbative regime. That is not relevant. The main thing is to be in the single exciton regime where $\langle N \rangle < 1$. There are many ways to guarantee this situation for fs excitation¹⁴, 21-23. But why switch to ps excitation. Moreover for pulsed excitation the unit should be fluence in $\mu\text{J} / \text{cm}^2$ not mW/cm^2 . It is a pulse with an energy not a power.
14. Fig3a shows the PLQY vs fluence for CW pumping. This is a fascinating result. It would be useful to consider the QD in terms of competition kinetics between k_{rad} and k_{nonrad} . Why not plot the k_{rad} and k_{nonrad} as a function of temperature since you have QD and k_{decay} . Similarly Fig3b shows the lifetime vs temperature. It should be the rate constant not the lifetime. Because the rate constant is linear in scale and the time constant is nonlinear. So the rate constant a better measure. Examining the rate constants vs temperature in an appropriate plot can be very revealing as shown here for unraveling excitonic spatial coherence¹⁵
15. Fig3c-d show the Intensity vs temperature $1/T$. This is a useful plot but it could be done better. The bright colors are distracting and not helpful to reading the data. I imagine the purpose of the temperature dependence is to see if there is some activation barrier to going from emitting states to non emitting states. Such a plot is best done as an Arrhenius plot¹⁵. The temperature dependence should also report on the energy shifting¹⁵ and bandwidth¹⁵, 24, 25 narrowing vs temperature to unravel details relevant to this situation.
16. The authors discuss the role of the surface and of surface ligands in controlling the PL kinetics. This point has been addressed in detail in CdSe QD²⁶⁻²⁸. The idea is that there are surface excitons that are shallow so are not populated at 300K. By 30K these shallow states become populated and emit. Their emission is broadened and redshifted from the exciton emission due to phonon progressions. By virtue of being at the interface of the QD core and the ligands, the interface is very sensitive to ligands for the PL behavior²⁹⁻³⁶.

Refs

- (1) Kambhampati, P. A Brief Discussion of Chemical Kinetics versus Chemical Dynamics. *The Journal of Physical Chemistry Letters* 2023, 14 (12), 2996-2999.
- (2) Strandell, D.; Wu, Y.; Mora-Perez, C.; Prezhdoo, O.; Kambhampati, P. Breaking the Condon Approximation for Light Emission from Metal Halide Perovskite Nanocrystals. *The Journal of Physical Chemistry Letters* 2023, 14, 11281-11285.
- (3) Shirasaki, Y.; Supran, G. J.; Bawendi, M. G.; Bulović, V. Emergence of colloidal quantum-dot light-emitting technologies. *Nature photonics* 2013, 7 (1), 13-23.
- (4) Ghosh, A.; Strandell, D. P.; Kambhampati, P. A spectroscopic overview of the differences between the absorbing states and the emitting states in semiconductor perovskite nanocrystals. *Nanoscale* 2023, 15 (6), 2470, 10.1039/D2NR05698D.

DOI: 10.1039/D2NR05698D.

- (5) Brosseau, P.; Ghosh, A.; Seiler, H.; Strandell, D.; Kambhampati, P. Exciton–polaron interactions in metal halide perovskite nanocrystals revealed via two-dimensional electronic spectroscopy. *The Journal of Chemical Physics* 2023, 159 (18).
- (6) Kambhampati, P. Learning about the Structural Dynamics of Semiconductor Perovskites from Electron Solvation Dynamics. *The Journal of Physical Chemistry C* 2021, 125 (43), 23571-23586. DOI: 10.1021/acs.jpcc.1c07445.
- (7) Seiler, H.; Palato, S.; Sonnichsen, C.; Baker, H.; Socie, E.; Strandell, D. P.; Kambhampati, P. Two-dimensional electronic spectroscopy reveals liquid-like lineshape dynamics in CsPbI₃ perovskite nanocrystals. *Nat. Commun.* 2019, 10, 4962.
- (8) Zhu, H. M.; Miyata, K.; Fu, Y. P.; Wang, J.; Joshi, P. P.; Niesner, D.; Williams, K. W.; Jin, S.; Zhu, X. Y. Screening in crystalline liquids protects energetic carriers in hybrid perovskites. *Science* 2016, 353 (6306), 1409.
- (9) Miyata, K.; Meggiolaro, D.; Tuan Trinh, M.; Joshi, P. P.; Mosconi, E.; Jones, S. C.; De Angelis, F.; Zhu, X. Y. Large Polarons in Lead Halide Perovskites. *Sci. Adv.* 2017, 3, e1701217.
- (10) Miyata, K.; Atallah, T. L.; Zhu, X. Y. Lead Halide Perovskites: Crystal-Liquid Duality, Phonon Glass Electron Crystals, and Large Polaron Formation. *Sci. Adv.* 2017, 3, e1701469.
- (11) Kambhampati, P. Unraveling the excitonics of light emission from metal-halide perovskite quantum dots. *Nanoscale* 2024, 16 (32), 15033-15058.
- (12) Strandell, D. P.; Zenatti, D.; Nagpal, P.; Ghosh, A.; Dirin, D. N.; Kovalenko, M. V.; Kambhampati, P. Hot Excitons Cool in Metal Halide Perovskite Nanocrystals as Fast as CdSe Nanocrystals. *ACS Nano* 2023.
- (13) Strandell, D. P.; Kambhampati, P. Light Emission from CsPbBr₃ Metal Halide Perovskite Nanocrystals Arises from Dual Emitting States with Distinct Lattice Couplings. *Nano Lett.* 2023, 23 (23), 11330-11336.
- (14) Strandell, D. P.; Kambhampati, P. Observing strongly confined multiexcitons in bulk-like CsPbBr₃ nanocrystals. *The Journal of Chemical Physics* 2023, 158 (15).
- (15) Strandell, D.; Wu, Y.; Prezhdo, O.; Kambhampati, P. Excitonic Quantum Coherence in Light Emission from CsPbBr₃ Metal-Halide Perovskite Nanocrystals. *Nano Lett.* 2023.
- (16) Strandell, D.; Dirin, D.; Zenatti, D.; Nagpal, P.; Ghosh, A.; Raino, G.; Kovalenko, M. V.; Kambhampati, P. Enhancing Multiexcitonic Emission in Metal-Halide Perovskites by Quantum Confinement. *ACS Nano* 2023.
- (17) Baker, H.; Strandell, D.; Kambhampati, P. Emitting State of Bulk CsPbBr₃ Perovskite Nanocrystals Reveals a Quantum-Confined Excitonic Structure. *J. Phys. Chem. C* 2020, 124 (34), 18816.
- (18) Kambhampati, P. Nanoparticles, Nanocrystals, and Quantum Dots: What Are the Implications of Size in Colloidal Nanoscale Materials? *J. Phys. Chem. Lett.* 2021, 12, 4769.
- (19) Li, B.; Brosseau, P. J.; Strandell, D. P.; Mack, T. G.; Kambhampati, P. Photophysical Action Spectra of Emission from Semiconductor Nanocrystals Reveal Violations to the Vavilov Rule Behavior from Hot Carrier Effects. *J. Phys. Chem. C* 2019, 123, 5092.
- (20) Mooney, J.; Krause, M. M.; Kambhampati, P. Connecting the Dots: The Kinetics and Thermodynamics of Hot, Cold, and Surface-Trapped Excitons in Semiconductor Nanocrystals. *J. Phys. Chem. C* 2014, 118, 7730.
- (21) Strandell, D. P.; Ghosh, A.; Zenatti, D.; Nagpal, P.; Kambhampati, P. Direct Observation of Higher Multiexciton Formation and Annihilation in CdSe Quantum Dots. *The Journal of Physical Chemistry Letters* 2023, 14 (30), 6904-6911.
- (22) Sonnichsen, C. D.; Strandell, D. P.; Brosseau, P. J.; Kambhampati, P. Polaronic quantum confinement in bulk CsPbBr₃ perovskite crystals revealed by state-resolved pump/probe spectroscopy. *Physical Review Research* 2021, 3 (2), 023147.
- (23) Palato, S.; Seiler, H.; Baker, H.; Sonnichsen, C.; Brosseau, P.; Kambhampati, P. Investigating the electronic structure of confined multiexcitons with nonlinear spectroscopies. *J. Chem. Phys.* 2020, 152 (10), 21, Article. DOI: 10.1063/1.5142180.
- (24) Mack, T. G.; Jethi, L.; Kambhampati, P. Strategy for Exploiting Self-Trapped Excitons in Semiconductor Nanocrystals for White Light Generation. *ACS Photonics* 2019, 6 (5), 1118-1124, Article. DOI: 10.1021/acsphotonics.9b00212.
- (25) Mack, T. G.; Jethi, L.; Kambhampati, P. Temperature Dependence of Emission Line Widths from Semiconductor Nanocrystals Reveals Vibronic Contributions to Line Broadening Processes. *J. Phys. Chem. C* 2017, 121, 28537.
- (26) Kambhampati, P. On the kinetics and thermodynamics of excitons at the surface of semiconductor nanocrystals: Are there surface excitons? *Chem. Phys.* 2015, 446, 92-107, Article. DOI: 10.1016/j.chemphys.2014.11.008.
- (27) Mooney, J.; Krause, M. M.; Saari, J. I.; Kambhampati, P. Challenge to the deep-trap model of the surface in semiconductor nanocrystals. *Phys. Rev. B* 2013, 87 (8), 5, Article. DOI: 10.1103/PhysRevB.87.081201.
- (28) Mooney, J.; Krause, M. M.; Saari, J. I.; Kambhampati, P. A microscopic picture of surface charge trapping in semiconductor nanocrystals. *J. Chem. Phys.* 2013, 138 (20), 9, Article. DOI: 10.1063/1.4807054.
- (29) Walsh, B. R.; Saari, J. I.; Krause, M. M.; Nick, R.; Coe-Sullivan, S.; Kambhampati, P. Surface and interface effects on non-radiative exciton recombination and relaxation dynamics in CdSe/Cd,Zn,S nanocrystals. *Chem. Phys.* 2016, 471, 11-17, Article. DOI: 10.1016/j.chemphys.2015.11.004.
- (30) Walsh, B. R.; Saari, J. I.; Krause, M. M.; Mack, T. G.; Nick, R.; Coe-Sullivan, S.; Kambhampati, P. Interfacial Electronic Structure in Graded Shell Nanocrystals Dictates Their Performance for Optical Gain. *J. Phys. Chem. C* 2016, 120 (34), 19409-19415, Article. DOI: 10.1021/acs.jpcc.6b05836.
- (31) Jethi, L.; Mack, T. G.; Krause, M. M.; Drake, S.; Kambhampati, P. The Effect of Exciton-Delocalizing Thiols on Intrinsic Dual Emitting Semiconductor Nanocrystals. *ChemPhysChem* 2016, 17 (5), 665-669, Article. DOI: 10.1002/cphc.201501049.
- (32) Walsh, B. R.; Saari, J. I.; Krause, M. M.; Nick, R.; Coe-Sullivan, S.; Kambhampati, P. Controlling the Surface of Semiconductor Nanocrystals for Efficient Light Emission from Single Excitons to Multiexcitons. *J. Phys. Chem. C* 2015, 119 (28), 16383-16389, Article. DOI: 10.1021/acs.jpcc.5b03853.
- (33) Krause, M. M.; Mack, T. G.; Jethi, L.; Moniodis, A.; Mooney, J. D.; Kambhampati, P. Unraveling photoluminescence quenching pathways in semiconductor nanocrystals. *Chem. Phys. Lett.* 2015, 633, 65-69, Article. DOI: 10.1016/j.cplett.2015.05.017.
- (34) Krause, M. M.; Kambhampati, P. Linking surface chemistry to optical properties of semiconductor nanocrystals. *Phys. Chem. Chem. Phys.* 2015, 17 (29), 18882-18894, Article. DOI: 10.1039/c5cp02173a.
- (35) Krause, M. M.; Mack, T.; Moniodis, A.; Mooney, J. D.; Kambhampati, P. Fluorescence quenching study of small colloidal nanocrystals yields insight into surface trapping processes. *Abstr. Pap. Am. Chem. Soc.* 2014, 248, 1, Meeting Abstract.

Reviewer #3

(Remarks to the Author)

This manuscript presents a novel and potentially impactful methodology for studying photoluminescence (PL) dynamics in perovskite quantum dot (PQD) films under operational conditions. The authors introduce an integrated "in-situ" spectroscopy setup that enables synchronous acquisition of steady-state and time-resolved PL (TRPL) under continuous-wave (CW) illumination, offering a more realistic picture of device-relevant light emission than conventional TRPL. This approach addresses a well-known limitation in the field, where ultrafast pulsed excitation used in TRPL experiments does not accurately reflect steady-state operational behavior. If validated and reproducible, this setup could become a valuable characterization tool for PQDs and other light-emitting materials.

The methodology is promising and generally sound. However, several points require clarification or additional evidence. First, the effect of CW intensity on the TRPL decay should be more rigorously assessed. The influence of laser pulse intensity is discussed, but a more detailed kinetic analysis would strengthen the conclusions. For example, the use of biexponential fitting alone may not fully capture the complexity of carrier dynamics, especially in systems where trap-assisted recombination or Auger processes may play a role. It would be beneficial to supplement the existing data with kinetic modeling to confirm that the derived time constants are consistent with the proposed mechanisms.

A key point requiring clarification is the role of the CW light in modifying the sample during measurement. The irreversible changes observed after pre-illumination (Figure 4) suggest that CW exposure alters the material in a non-trivial way. This raises concerns about whether the in-situ TRPL truly reflects steady-state behavior, or instead captures a transient state of a material in flux. Can the authors demonstrate that measurements are reproducible on the same sample, and between different samples prepared under identical conditions? This would bolster the validity of the method and the conclusions drawn.

Further experimental details would also enhance reproducibility and interpretation. The authors should provide the repetition rate of the pulsed laser and explain how they confirmed that charge accumulation is avoided between pulses. Figure S8 is used to argue that the pulsed beam does not affect the observed kinetics; however, time constant values are not provided, and should be included to support this claim.

A few specific points are noted below:

Small changes:

- Figure 2a and line 90
- o Figure says unannealed, while line 90 says annealed.
- Figure 4. b) and d) is practically impossible to distinguish between the solid line and the dotted line. Change colour scheme.
- Line 130, accompanied.

Questions about the paper:

1.- For Figure 2, the conditions (pump intensities) for CW and TRPL measurement are quite different, which implies different charge carrier generation and recombination rates (DOI: 10.1063/1.5143121) and can potentially explain differences in the trends presented in the paper. On the other hand, "in-situ TRPL" measurements employ CW illumination as a base, which they show can affect the PL response of the system over time (Figure 4). If the sample changes over time due to the CW, how can we rely on the "in-situ" TRPL information obtained? Can the authors show that the "in-situ" measurements are reproducible? Can the measurements be repeated in the same system already measured and obtain the same results? If not, can the experiments be repeated and obtain the same behaviour?

2.- The authors claim the following (line 105):

"According to what has been extensively documented in previous studies, the emergence of a rapid PL decay component at high pump fluence is attributed to the Auger recombination of multi-excitons, which, however, is ruled out due to the ultralow intensity of the pulsed laser in our case."

And use Figure S8 to show that the kinetics do not change by changing the intensity of the pulsed laser, but 2 questions emerge from this statement:

a) The authors employ high CW pump intensities. The power of illumination to excite a sample in a TRPL experiment has a strong influence on the concentration of photogenerated carriers, which in turn determines the type of recombination process that dominates in the system, including Auger recombination for high fluences ((DOI: 10.1039/d0c904950f), (DOI: 10.1063/1.5143121), (DOI: 10.1002/adfm.201910004). Have authors consider Auger recombination as a result of the CW illumination in the "in-situ" TRPL, or what was the criteria to discard this phenomenon as a result of the CW illumination?

b) For Figure S8, the authors make the following statement: "The intensity of the pulsed beam has nothing to do with the observed in-situ TRPL kinetics in our case...". However, it seems they do not provide time constant values for figure S8 that can corroborate this statement. I would recommend including this information.

3.- The authors are using a sum of exponentials to obtain 2 different time constants and then obtain an average value of the lifetime. What does this lifetime represent?

4.- It has been reported that the laser intensity influences the TRPL decay and the dominant recombination mechanisms, but this information is often hard to evaluate from a biexponential fitting (DOI: 10.1039/d0c904950f). Have the authors consider using a kinetic model to analyse the data and corroborate that the same conclusions can be reached? (DOI: 10.1002/adfm.201910004) (DOI: 10.1039/D0CP04950F)

5.- Could the authors provide the repetition rate employed for the laser? How did the authors corroborate that the repetition rate was adequate for the system, particularly at preventing charge accumulation in the material? (DOI: 10.1039/d0c904950f)
6.- As mentioned in Figure 4, the samples seem to change under CW conditions over time, which could affect the results of "in-situ" TRPL. Could the authors comment on the influence of the CW on "in-situ" TRPL regarding sample relaxation/charge accumulation between laser pulses?

I feel like if the points are covered in good detail and a robust nature then I would be considered for publication but currently would recommend these revisions be made prior to publication.

Reviewer #4

(Remarks to the Author)

Version 1:

Reviewer comments:

Reviewer #1

(Remarks to the Author)

The revised version of the manuscript addresses many of the issues raised in the initial round of review, both by myself and by the other reviewers. The data are now presented more consistently, and the discussion is better structured. However, I must say that the revised paper feels like an entirely different manuscript. A major part of the original work — the time evolution of the device — has now been removed. As a result, the focus and scope have shifted significantly.

I still find the distinction between what the authors call "operando" and "non-operando" data rather unclear. In Figures 4 and 5, the two datasets are now presented on identical scales, which is good. However, they are background-corrected differently, and the operando data clearly contain dark counts not present in the non-operando measurements. This background difference — particularly the dark counts around negative times — appears to be the main source of the observed changes in decay behavior.

Overall, the differences between operational and non-operational conditions have become quite small. Yet, due to inconsistent background levels across datasets and a lack of rigorous analysis of this background, it remains extremely difficult to quantify these differences in any meaningful way.

Fundamentally, I struggle to see what new insight or capability the technique actually provides. The results are not convincingly quantified, and in the current form I cannot recommend the manuscript for publication. Even if the authors were to conduct a more robust background analysis and present more convincing data, I am not convinced the changes would be significant enough to yield meaningful conclusions about the material's behavior — certainly not at a level suitable for Nature Communications.

Reviewer #2

(Remarks to the Author)

The authors have well revised the manuscript and answered the questions. This paper is suitable for publication with a minor addition.

Since the authors are discussing TRPL with a streak camera on LHP QD< I encourage them to cite the most recent and precise work on the subject published by *Wlopw*.

<https://pubs.rsc.org/en/content/articlelanding/2024/nr/d4nr01481b/unauth>

<https://pubs.aip.org/aip/jcp/article-abstract/158/15/154702/2882245/Observing-strongly-confined-multiexcitons-in-bulk?redirectedFrom=fulltext>

<https://pubs.acs.org/doi/abs/10.1021/acs.nanolett.3c03975>

<https://pubs.acs.org/doi/abs/10.1021/acs.nanolett.3c03180>

Reviewer #3

(Remarks to the Author)

I am satisfied that the authors have fully addressed the concerns raised in the initial review. I believe the authors have addressed the concerns of the reviewers generally very well. The revisions provide clear experimental details, improved data presentation, and quantitative analysis that support the conclusions. The rationale for the "operando" CW-bias TRPL

approach is better articulated - I believe it is potentially useful for the authors to consider adding a one-sentence definition early in the Introduction clarifying the use of operandi to mean under steady-state CW optical conditions, emulating device illumination conditions, rather than any electrical bias.

The manuscript is consistent and much improved from the original manuscript - I believe this will be of benefit to the scientific field and nor warrants publication subject to perhaps some minor editorial corrections for clarity and consistency.

Reviewer #4

(Remarks to the Author)

Version 2:

Reviewer comments:

Reviewer #1

(Remarks to the Author)

I think the manuscript improved overall and the presentation is better. I can follow things now, and I find that the main point — that the trend is superlinearity under CW light and sublinearity under pulsed excitation — is clear. I think the authors convinced me on that point, and I agree this is probably a reflection of trapping, and that this approach can be used to study trapping. In a very stretched way, it can also be related to “operando” conditions, though that’s limited to optical illumination, not a real device bias... I personally would not call it operando and most device people as well would be confused.

The part about lifetime change is still shaky. The graph itself doesn’t convince me. The authors really try to sell it, but the experiments are so different — illumination power, laser system, repetition rate, everything. You can’t directly compare absolute lifetimes between these two regimes. It’s okay to show this data, but it shouldn’t be pushed as a key point.

I was also a bit hesitant about data processing, but I think that even if there are some issues there, the main message is still sound.

Based on this, I think the manuscript is generally publishable. It shows a certain level of robustness, especially if the authors are more honest about what is clearly demonstrated and what is more speculative.

But does it fit Nature Communications? I don’t think so. The authors show that under certain measurement conditions you can see effects of trapping. That’s good, but it’s not groundbreaking. From this point of view, it’s probably not for Nature Communications. The quantitative part also doesn’t feel robust enough, that’s my honest opinion.

Finally, I know the other two reviewers were quite positive, but I don’t feel they looked deep enough into the data to really see these methodological nuances and what actually matters here. If the editor has doubts, I would suggest asking for the opinion of another spectroscopist, someone who really works on photochemical kinetics. I think that would help to evaluate the impact properly. In my view, the impact doesn’t fit Nature Communications. Scientific Reports or Communications Chemistry would be a more suitable.

Response to the editor's and reviewers' comments:

Response to the editor's comments

In particular, the referees collectively raise critical concerns influence of the CW light in modifying the sample during the measurements, making the influence of photodegradation effects unclear, as well as whether the current approach provides operando data and questioning the reproducibility of the experiments. In addition, reviewer #1 is concerned with data presentation, potentially exaggerating difference of in-situ TRPL to conventional TRPL. Furthermore, reviewer #2 is concerned with the choice of the system and comparison of pristine and annealed films.

Response:

We are grateful to the editor and reviewers for pointing out the issues of CW light-induced photodegradation, experimental reproducibility, data presentation, and the study systems, which may pose potential problems in exploring the intrinsic operando behavior of charge carriers.

- (1) Photodegradation and reproducibility issues. To avoid the occurrence of irreversible photodegradation and ensure experimental reproducibility, we fabricate photostable PNC film via a successive thermal annealing and CW preillumination treatment. The presented results (e.g., **Figure 2f**, **Figure 3b**, **Figure 3c**, **Figure S7**, and **Figure S8**) collectively verify the suppression of photodegradation for the tested perovskite nanocrystal (PNC) films during spectroscopic measurements. Furthermore, a series of steady-state and time-resolved confirmatory experiments have been added to verify the reproducibility (**Figure S13** and **Figure S14**). More detailed discussion will be shown later to address the reviewers' concerns.
- (2) Data presentation issue. The presentation of TRPL data has been improved according to Reviewer-1's suggestion. As shown in **Figure 4c**, **Figure 4d**, **Figure 5a**, and **Figure 5b**, all the typical and in-situ TRPL data share the identical vertical and horizontal scales. Upon these revisions, the different features in TRPL kinetics are more clearly recognized now.
- (3) Research system issue. As stated above, we have replaced the previous research system (e.g., the unstable PNC films with or without an annealing treatment) with the improved, ultrastable PNC film in the revised manuscript. Per the comment of Reviewer-2, the manuscript has been modified to concentrate on the operando kinetics of charge recombination in stable PNC film. The comparison between the unannealed and annealed films, in which irreversible photodegradation occurs during the optical experiments, has been removed from the manuscript. Such a revision engenders a more compact logic structure and a more reliable conclusion.

Response to the Reviewer-1's comments

The paper by Cao et al. presents a spectroscopic study of quantum dot perovskite materials, with a focus on their potential use in color conversion layers. The authors perform time-correlated single-photon counting (TCSPC) measurements under steady-state illumination bias and conclude that multiple processes are involved in controlling the emission dynamics, including Auger recombination and photodegradation-related photochemistry.

The study gives me mixed feelings. On one hand, the general approach is interesting. Performing measurements under steady-state illumination bias is a promising way to understand the dynamics of materials under conditions relevant to actual device operation. On the other hand, instead of carefully presenting and analyzing their results, the authors often push forward broad, principle-challenging claims and present their data in a way that can be misleading. I find that the paper, in its current form, is not methodologically solid and would require substantial revision before it could be considered for publication. Furthermore, I feel the work does not meet the high standards expected for Nature Communications, and would be more suitable for a specialized journal such as Chemistry of Materials or Communications Chemistry.

1. The authors describe their measurements as "operando dynamics." I believe this is misleading. The term would be appropriate if the processes they observed were reversible, and if steady-state illumination accurately simulated device operating conditions. However, many of the changes observed are irreversible, involving photodegradation and permanent chemical alterations. Therefore, the premise that the study represents true "operando" conditions is flawed. A major part of the introduction and context needs to be rethought and rewritten to reflect that the focus should be on photodegradation, not just dynamic processes during operation.

Response:

We thank the reviewer for highlighting the potential issues that may have arisen due to the irreversible alterations undergone during the CW measurements. Following a more exhaustive examination, it has been determined that the irreversible PL enhancement/decline, as well as the lifetime prolongation/shortening, should be ascribed to the chemical instability of the perovskite nanocrystal (PNC) films rather than the operando properties demonstrated in the previous manuscript. In the revised manuscript, we have redesigned the tested sample and conducted a series of experiments to verify the experiment reproducibility and the capability of the integrated in-situ PL platform in tracking operando TRPL kinetics, as detailed below:

- (1) Ultrastable PNC films are prepared through a two-step post-synthesis treatment method, comprising annealing and preillumination. Thermal annealing is suggested to remove excess organic ligand, thus preventing the occurrence of irreversible PL decline related to photo-induced ligand detachment of the dispersed perovskite quantum dots (Figure 4a of the previous manuscript). Concurrently, the preillumination treatment can improve both the

PLQY and stability of the sintered PNC films in terms of defect self-healing, thereby circumventing the irreversible PL enhancement of the annealed PNC film (Figure 4c of the previous manuscript).

Figure R1. PL intensity evolution of the PNC films with different treatments under continuous light illumination (corresponding to Figure 2f).

As verified by in-situ PL spectroscopy and summarized in **Figure R1**, the two-step treated film possesses excellent photostability for subsequent optical characterizations.

- (2) To guarantee that all the experimental findings are reversible and reproducible, we have carefully controlled the CW light intensity to be lower than 15 mW/cm².

Figure R2. Steady-state PL spectra (left) and TRPL kinetics (right) of the PNC film before and after a complete experimental period (corresponding to Figure S13 and Figure S14, respectively).

Furthermore, both the steady-state PL spectra and TRPL kinetics for the same PNC film before and after a complete experimental period have been measured for quantitative comparison (**Figure R2**). The variations in PL intensity and TRPL lifetime are less than 5%, in great line with the robustness of the improved sample in spectroscopy measurements.

Accordingly, the manuscript has been thoroughly revised. Firstly, additional context was incorporated to introduce the two-step treatment method and to highlight the resultant variation of photophysical properties. Secondly, more details of the operando PL measurements have been added in the manuscript, which demonstrates the stability of both the steady-state PL and TRPL units of the testing platform. Thirdly, as displayed above, the spectroscopic features are carefully compared before and after the entire experimental period, excluding the impact of irreversible factors (*e.g.*, undesired photodegradation) on the PL features observed in the present work.

Revisions:

Page 4: *The films are fabricated by spin-coating the purified PNC colloidal solutions onto a quartz substrate subsequently followed by thermal annealing and preillumination treatments, as schematically illustrated in Figure 2a.*

Page 4: *Figure 2f displays the PL intensity evolution versus illumination time for the three types of PNC films, as derived from the long-term in-situ PL spectroscopy measurement (Figure S7), which unambiguously manifests the efficacy of the sequential annealing and preillumination treatment on optimizing the photostability.*

Page 5: *Figure 3b displays the as-obtained in-situ PL spectra, which are centered at 535 nm regardless of illumination period, aligning closely with the standard PL spectrum shown in Figure 2e. The fluctuation of PL intensity is found to be less than 2% over the entire experimental period (Figure S8), validating the prominent stability of the integrated PL spectroscopy platform.*

Page 5: *Concomitant with the in-situ PL spectra measurement, TRPL results are obtained by means of TCSPC. Three sets of TRPL kinetics with the same accumulation time of ~660 s, corresponding to the three regions denoted in in-situ PL spectra as exemplified in Figure 3c, share the identical time-resolved features.....Secondly, the three sets of TRPL kinetics are self-similar with a variation of average lifetime as small as ~4% (Figure S9 and Table S1), indicative of excellent reproducibility for the in-situ TRPL test.*

2. I am very disappointed by the presentation of Figures 2c, d, g, and h. It appears that the figures were designed in a way that could mislead an uncareful reader. In comparing conventional TRPL and photoluminescence under illumination bias, the data should be presented on identical scales, and the excitation intensities should be normalized considering the excitation density in the sample. The authors fail to do this, instead presenting the datasets at very different scales and intensities to suggest that the results with and without bias are dramatically different — which is not true. After extracting and replotting the data myself, I find that the kinetics are actually very similar; only the trends with increasing illumination intensity differ somewhat. This is a serious flaw in data presentation.

Response:

We thank the reviewer for the important comments on data presentation. In the revised manuscript, we have normalized scales and intensity scales for the typical TRPL and in-situ TRPL of the PNCs, as suggested by the reviewer. **Figure 4c, Figure 4d, Figure 5a, and Figure 5b** now clearly show the difference between the typical and in-situ TRPL kinetics and that between the PNC solution and PNC film.

Regarding the excitation density, it seems insignificant to normalize the values of the pulse beam in typical TCSPC measurements and the CW beam in in-situ TCSPC measurements. For example, as stated in the revised manuscript, the peak power density of the fs pulse is several orders of magnitude stronger than that of the CW light, when their average pump fluences are comparable. Specifically, the pump power of the pulse laser strongly depends on its repetition rate. Therefore, as Reviewer-2 pointed out, to circumvent this issue, the pump fluence of the pulse beam should be shown in units of “ $\mu\text{J}/\text{cm}^2$ ” instead of “ mW/cm^2 ”, which has been done in the present manuscript.

Revisions:

Page 6 & Page 7:

Page 2: *At a comparable pump fluence, for instance, the peak power density of the ultrafast pulse is several orders of magnitude stronger than that of CW light. Therefore, a series of nonlinear optical processes can be detected by TRPL, while their occurrence is much less probable in real devices.*

3. Figure 4 clearly shows the effects of photodegradation and related permanent photochemical changes. However, the authors do not attempt to quantify the contribution of these irreversible changes compared to other dynamic effects such as multiparticle interactions and Auger processes. Without a quantitative analysis, it is very difficult to judge how significant the reported findings are. A thorough decomposition and estimation of different contributions are essential.

Response:

We appreciate the reviewer for reminding us of considering the influence of photodegradation and related permanent photochemical changes on the observed spectroscopic variations. As replied to the first comment, we have made substantial efforts to improve the stability of the tested sample and to validate the reliability of the in-situ spectroscopy platform, which are displayed in **Figure R1** and **Figure R2**, respectively. The experimental results collectively verify that photodegradation as well as any irreversible alterations are completely suppressed in the revised manuscript.

Having excluded the interference of permanent photochemical changes, we further perform systematic quantitative analysis (**Figure 5a-5c**, **Figure S15**, **Figure S16**, **Table S2**), accompanied by optical imaging characterizations (**Figure 5d**), to rationalize the specific in-situ TRPL features that substantially differ from the typical TRPL data. This ultimately allows us to develop an accurate physical model (**Figure 5f**) derived from the equilibrium-state classical continuity equation, which excellently interprets the landscape of charge recombination kinetics in PNC films under operational conditions.

Revisions:

Page 4: *The films are fabricated by spin-coating the purified PNC colloidal solutions onto a quartz substrate subsequently followed by thermal annealing and preillumination treatments, as schematically illustrated in Figure 2a.*

Page 4: *Figure 2f displays the PL intensity evolution versus illumination time for the three types of PNC films, as derived from the long-term in-situ PL spectroscopy measurement (Figure S7), which unambiguously manifests the efficacy of the sequential annealing and preillumination treatment on optimizing the photostability.*

Page 7: *As illustrated in Figure 5a, the in-situ TRPL kinetics of the PNC solution present negligible alteration in response to a wide range of CW pump intensity variations. No additional rapid decay component is observed at high pump fluence, in contrast to the typical TRPL data shown in Figure 1c, indicating that the multiexciton effect is minimized under CW photoexcitation. The distinctive in-situ TRPL behaviors facilitate the ascription of the retarded TRPL decay to the specific microscopic structure of PNC films, whose formation is accompanied by the transition from highly dispersed-state nanoparticles in solution (Figure 1a) to sintered close-packed large grains in solid-state configuration (Figure 2c).*

Page 7: In quantitative, provided that multi-body Auger recombination is negligible, the equilibrium-state classical continuity equation in polycrystalline perovskites is typically expressed as: $dn_0/dt = G - k_1n_0 - k_2n_0^2 = 0$, where n_0 is the carrier density at the equilibrium state, t is time, G is the rate of photocarrier generation, k_1 is the rate constant for first-order nonradiative recombination, and k_2 is the rate constant for bimolecular recombination. Once a perturbation is superposed on the equilibrium state, the expression should be rewritten as: $d(n_0 + \delta n)/dt = G - k_1(n_0 + \delta n) - k_2(n_0 + \delta n)^2$, in which δn is the density of excess carrier with a value much smaller than equilibrium-state carrier density in a perturbation case ($\delta n \ll n_0$). A combination of the two aforementioned equations results in the classical continuity equation being reduced to the following form: $d\delta n/dt = -k_1\delta n - 2k_2n_0\delta n$, which is indicative of first-order reaction kinetics. In-situ TRPL experiment fulfills the prerequisites of perturbation, therefore the time-dependent PL intensity is predicted to be described by the first-order kinetics model with a decay lifetime of τ , $I_{PL}(t) = I_{PL}(0) \cdot \exp(-t/\tau)$. Furthermore, in consideration of the inhomogeneity of the chemical environment for individual perovskite grains, it has been proposed that the TRPL data be fitted more accurately with a bi-exponential function²⁴ or a stretched exponential function²⁵. Here we adopt the bi-exponential model for elucidation due to its higher fitting goodness than the stretched exponential model (Figure S15).

Page 7: A representative in-situ TRPL data in conjunction with the biexponential fitting results are depicted in Figure 5b, where the fast (τ_1) and slow (τ_2) decay components have been suggested to arise from charge recombination at the grain interface/surface and within the volume of grains, respectively^{26,27}.

Page 8: By visualizing the light emission patterns, wide-field PL microscopy images (Figure 5d) further verify the improved PL efficiency of PNC films upon a strong CW illumination. Specifically, the PL imaging experiments are conducted by simultaneously reducing the excitation power (P) and prolonging the integrating time (Γ) with the same factor. While the product of P and Γ remains constant, the PL intensity exhibits a prominent decrease with the reduction in P (see detailed analysis in Figure S17), indicative of a higher PLQY by enhancing the CW illumination.

4. Similarly, Figure 5 is highly qualitative and does not allow the reader to assess the confidence level of the conclusions or sufficiently support the interpretation of the spectroscopic data. Figure 5 is just a cartoon, not adding much to the study or supporting it. A more rigorous, controlled investigation correlating structural changes (upon annealing or in some other way) to optical properties, using systematical and quantitatively well-defined sample sets, would be needed to substantiate the proposed interpretations. As it stands, the data do not have the level of rigor expected for a Nature Communications publication.

Response:

We thank the reviewer for indicating the issue of qualitative analysis in the previous manuscript, which looks insufficiently rigorous for supporting the physical model. In order to address this concern, **Figure 5** has been carefully revised from four aspects as detailed below:

- (1) Quantitative analysis. Within the framework of classical continuity equation in perovskite films, we have mathematically proved that charge recombination kinetics in a perturbation manner follows the first-order reaction description (**Page 7**). This is premised on the fact, $\delta n \ll n_0$, as has been manifested to be held for the in-situ TRPL measurement (**Page 5**, **Figure S10**, **Figure S11**). In addition, considering the inhomogeneity of individual perovskite grains, the ensemble TRPL behavior generally turns out to be a multi-exponential or stretched exponential form. In line with the suggestion of Reviewer-2, we select the bi-exponential function to fit the TRPL kinetics due to its higher fitting goodness. (**Figure 5b**, **Figure 5c**, **Figure S16**, **Table S2**). The quantitative analysis results help highlight the significant role of the charge-carrier and trap-state interaction in determining the TRPL features.
- (2) Comparison of PNC with different dispersion states. As discussed in the manuscript (**Page 7**), the retarded TRPL decay with the increase of CW light intensity arises from the PNC sintering. Such phenomenon observed in close-packed PNC film is absent in PNC colloidal solutions under the identical experimental conditions (**Figure 5a**), providing an additional support to the proposed physical model.
- (3) Additional visualization evidence. In order to more clearly showcase the CW light-induced brightening of PNC films, wide-field PL imaging test was conducted for visualizing this effect (**Figure 5d**). The CW light intensity and the integrating time are simultaneously changed to keep their product unchanged. If Auger recombination occurs at high CW intensities, the PL image will become dark with the elevation of light intensity and shortening of integrating time; otherwise, in the case of linear PL, the image brightness is anticipated to be unchanged. Interestingly, we have unambiguously observed an enhanced brightness under the strong CW illumination, which differs from the predication of both Auger recombination and linear PL and is in excellent agreement with reversable charge-carrier and trap-state interaction stated in our work.
- (4) Refined physical model. In the revised manuscript, the physical model has been improved by connecting the quantitative analysis with a more comprehensive consideration of the

PNC structure evolution (Figure 5e) and the steady-state/time-resolved spectroscopy results. Moreover, the physical symbols used in the mathematic equations are clearly denoted in the schematic (Figure 5f), which explicitly uncover the effect of CW light on increasing the PL efficiency. With these improvements, we believe that the physical model is more rigorous now and provides sufficient information to interpret the experimental findings.

Revisions:

Page 7: *As illustrated in Figure 5a, the in-situ TRPL kinetics of the PNC solution present negligible alteration in response to a wide range of CW pump intensity variations. No additional rapid decay component is observed at high pump fluence, in contrast to the typical TRPL data shown in Figure 1c, indicating that the multiexciton effect is minimized under CW photoexcitation. The distinctive in-situ TRPL behaviors facilitate the ascription of the retarded TRPL decay to the specific microscopic structure of PNC films, whose formation is accompanied by the transition from highly dispersed-state nanoparticles in solution (Figure 1a) to sintered close-packed large grains in solid-state configuration (Figure 2c).*

Page 7: *In quantitative, provided that multi-body Auger recombination is negligible, the equilibrium-state classical continuity equation in polycrystalline perovskites is typically expressed as: $dn_0/dt = G - k_1n_0 - k_2n_0^2 = 0$, where n_0 is the carrier density at the equilibrium state, t is time, G is the rate of photocarrier generation, k_1 is the rate constant for first-order nonradiative recombination, and k_2 is the rate constant for bimolecular recombination. Once a perturbation is superposed on the equilibrium state, the expression should be rewritten as: $d(n_0 + \delta n)/dt = G - k_1(n_0 + \delta n) - k_2(n_0 + \delta n)^2$, in which δn is the density of excess carrier with a value much smaller than equilibrium-state carrier density in a perturbation case ($\delta n \ll n_0$). A combination of the two aforementioned equations results in the classical continuity equation being reduced to the following form: $d\delta n/dt = -k_1\delta n - 2k_2n_0\delta n$, which is indicative of first-order reaction kinetics. In-situ TRPL experiment fulfills the prerequisites of perturbation, therefore the time-dependent PL intensity is predicted to be described by the first-order kinetics model with a decay lifetime of τ , $I_{PL}(t) = I_{PL}(0) \cdot \exp(-t/\tau)$. Furthermore, in consideration of the inhomogeneity of the chemical environment for individual perovskite grains, it has been proposed that the TRPL data be fitted more accurately with a bi-exponential function²⁴ or a stretched exponential function²⁵. Here we adopt the bi-exponential model for elucidation due to its higher fitting goodness than the stretched exponential model (Figure S15).*

Page 7: *A representative in-situ TRPL data in conjunction with the biexponential fitting results are depicted in Figure 5b, where the fast (τ_1) and slow (τ_2) decay components have been suggested to arise from charge recombination at the grain interface/surface and within the volume of grains, respectively^{26,27}.*

Page 8: *By visualizing the light emission patterns, wide-field PL microscopy images (Figure 5d) further verify the improved PL efficiency of PNC films upon a strong CW illumination. Specifically, the PL imaging experiments are conducted by simultaneously*

reducing the excitation power (P) and prolonging the integrating time (T) with the same factor. While the product of P and T remains constant, the PL intensity exhibits a prominent decrease with the reduction in P (see detailed analysis in Figure S17), indicative of a higher PLQY by enhancing the CW illumination.

Page 8: The landscape of operando charge recombination revealed by in-situ TRPL is illustrated in Figure 5f, where $n_{0,i}$ and δn denote the charge carriers generated by the CW beam and the pulse beam, satisfying the perturbation requirement, $\delta n \ll n_{0,i}$. The decay of δn involves two pathways: trap state-mediated nonradiative charge recombination (dashed arrow) and radiative charge recombination (solid arrow), which correspond to the first and second terms, respectively, in the continuity equation $d\delta n/dt = -k_1\delta n - 2k_2n_0\delta n$. With the elevation of the CW pump intensity, the value of n_0 is increased, which is anticipated to accelerate the TRPL decay; by contrast, the density of empty trap states (i.e., nonradiative recombination centers) is decreased owing to the trap filling by charge carriers, thus resulting in a reduced k_1 and a retarded recombination kinetics. Under operational conditions, the density of charge carriers is several orders of magnitude smaller than the trap state density³²⁻³⁵. This indicates that the effect of photoexcitation on filling trap states is dramatically greater than that on increasing n_0 . As a consequence, the in-situ spectroscopy results demonstrate a concomitant increase in the steady-state PL intensity and prolongation in the TRPL lifetime with the enhanced CW pump fluence, which mirrors the landscape of charge recombination in PNC films under realistic operational conditions that is not revealed by typical TRPL techniques.

Response to the Reviewer-2's comments

This manuscript reports on the spectroscopy of lead-halide perovskite quantum dots (QD) under operational conditions for LEDs. This is a very interesting and valuable manuscript and probably should be published. But there are many questions to address.

1. The authors use PL dynamics initially and then switch to kinetics. The initial point is that consistency in terms should be used. Pick one or the other. But it turns out that that the semantics matter. PL rarely measures dynamics. It mostly measures kinetics. There is a recent discussion of the differences between kinetics and dynamics. And there are recent examples of t-PL dynamics in LHP QD.

Response:

We thank the reviewer for highlighting the difference between dynamics and kinetics. As mentioned by the reviewer, the TCSPC data that display the PL decay profile as a result of charge recombination should be labelled “*kinetics*” rather than “*dynamics*”. The term has therefore been unified to “*kinetics*” in the revised manuscript.

Revisions:

Page 1, Title: *Operando Recombination Kinetics in Perovskite Nanocrystal Films Revealed by In-Situ Time-Resolved Photoluminescence.*

Page 1, Abstract: *In contrast to the established Auger recombination model, which suggests a more rapid TRPL kinetics at high pump fluence, it is demonstrated that the dependence of CW-driven PL intensity on pump fluence is governed by charge recombination kinetics in the context of charge-carrier and trap-state interactions.*

Page 2: *The sophisticated design of the photoexcitation method enables precise examination of the exciton/charge recombination kinetics of PNCs under the sustained excitation of a CW beam. As a proof of concept, the TRPL kinetics of photostable, close-packed PNC films were investigated by varying the CW excitation intensity over twenty folds.*

Page 5: *The integrated in-situ PL spectroscopy technique proposed for interrogating the operando charge recombination kinetics of PNC films is presented in Figure 3a.*

Page 6: *Having experimentally verified the reliability of the integrated in-situ PL spectroscopy platform, we conduct a systematic exploration into the operando charge recombination kinetics of PNC films.*

Page 8: *With the elevation of the CW pump intensity, the value of n_0 is increased, which is anticipated to accelerate the TRPL decay; by contrast, the density of empty trap states (i.e., nonradiative recombination centers) is decreased owing to the trap filling by charge carriers, thus resulting in a reduced k_1 and a retarded recombination kinetics.*

2. The introduction begins with the good features of LHP QD for LED. But why not CdSe and other covalent QD that were investigated first? Why are LHP QD special. One answer is their unique lattice that is defect tolerant and does so by supporting liquid-like structural dynamics.

Response:

We thank the reviewer for the suggestion of emphasizing the motivation behind the investigation of the perovskite nanocrystals (PNCs) and the recommendation of important references.

As presented in the revised introduction of the manuscript, one of the most important reasons for selecting PNC, in agreement with the reviewer's comment, is the unique physicochemical properties of PNCs, such as the significant defect tolerance (A. L. Rogach, *et al. ACS Energy Lett.*, **2017**, *2*, 2071) and polaronic features (P. Kambhampati, *et al. Nat. Commun.*, **2019**, *10*, 4962), differing from the traditional binary QDs. In the meantime, in contrast to the rapid development of high-efficiency PNC optoelectronic devices, mechanistic studies on the operando charge carrier kinetics still lag behind, which serves as another reason for delving into the operando behaviors of PNCs.

The relevant contexts and references have been added to the revised manuscript.

Revisions:

Page 2: *In addition to the significant advances in material design and device fabrication, considerable efforts have been also made over the past decade to elucidate the unique physicochemical properties of PNCs, such as excellent defect tolerance⁷ and drastic polaronic features⁸, aiming to further extend the utilization of PNCs in specific practical applications.*

3. The authors refer to t-PL measurements with streak cameras. They should cite the most recent and most detailed t-PL measurements of LHP QD that reveal their dynamics and kinetics for particles in dispersion.

Response:

We thank the reviewer for this constructive suggestion. Accordingly, one of the recently published references recommended by the reviewer (P. Kambhampati, *et al. Nano Lett.*, **2024**, 24, 61) has been cited to replace the previous one in the revised manuscript.

Revisions:

Page 2: *Time-resolved PL (TRPL) techniques, including streak camera⁹, fluorescence upconversion¹⁰, and time-correlated single photon counting (TCSPC)¹¹ are hitherto the most widely used approaches for investigating the energy conversion mechanism of PNCs.*

4. My main concern is that the annealed QD film is so different from the pristine film that it may not be meaningful to compare. The pristine film contains QD of 10 nm diameter. These are true QD. But the annealed film contains nanocrystals of 100 nm diameter which are not QD. Some NC are QD and some QD are NC but they are not the same as recently reviewed. The main issue is the diameter relative to the exciton Bohr length which confers QD effects to excitonic structure and dynamics.

Response:

We thank the reviewer for pointing out the misuse of “QD” in this work and the potential confusion it may have caused. As replied to the comments from Reviewer-1, the revised manuscript now focuses on the photostable, close-packed PNC film.

The comparison between the PQD solution and the PNC film has been removed due to the instability issue of the former during optical experiments, which causes irreversible photodegradation. In addition, the annealing time has been prolonged to 30 minutes to yield a much more homogeneous and sintered PNC film with an apparent size far beyond the exciton Bohr diameter. This minimizes the influence of the quantum confinement effect on PL kinetics and makes the landscape of charge recombination analogous to that of polycrystalline perovskite films. To eliminate any confusion, we have replaced the term “perovskite quantum dot (PQD)” with “perovskite nanocrystal (PNC)”.

Revisions:

The *title, abstract, and main text* have been thoroughly revised in line with the reviewer’s suggestion.

5. Fig1g shows the Abs spectra of the two films which look totally different. Why is there a more prominent excitonic peak for the annealed QD that increase in size to NC that are not QD.

Response:

The enhanced excitonic absorption peak accompanied by a redshift of the Abs spectra upon thermal annealing has been widely observed in PNC films (Kamat *et al.*, *ACS Mater. Lett.*, **2019**, *1*, 8; *J. Am. Chem. Soc.*, **2016**, *138*, 8603; *Acc. Chem. Res.*, **2021**, *54*, 520).

Figure R3. UV-vis absorption spectra of CsPbBr₃ nanocrystal films with different thermal annealing periods (Reprinted with permission from *et al.*, *ACS Mater. Lett.*, **2019**, *1*, 8. Copyright 2019 American Chemical Society.). With the prolongation of annealing time from 0 min to 60 min, the excitonic peak gradually becomes more prominent and shifts to the long-wavelength side.

As exemplified in **Figure R3**, this phenomenon has been suggested to be an indication of nanoparticle sintering (“Specifically, the appearance of a sharper excitonic peak in the bulk samples indicates the growth of very homogeneously sized large particles compared to the ensemble of small particles in the NC film samples.” This text is cited from Kamat *et al.*, *ACS Mater. Lett.*, **2019**, *1*, 8). This interpretation seems consistent with the experimental data reported and presented.

6. Typo errors here: “Now we move forward to the PL mechanism of PQD films under the operational conditions of CCLs. Figure 2a displays 90 the steady-state PL emission spectra of the annealed film excited by a CW beam” . It should be unannealed film

Response:

We thank the reviewer for pointing out the typo errors. The manuscript has been thoroughly reorganized and carefully revised, and we that all the typos have been calibrated in the revised version.

Revisions:

The *title, abstract, and main text* have been thoroughly revised.

7. Fig2b. Why does the pristine film show a sublinear saturation curve for intensity? It should slow slightly due to XX formation. It may also slow due to hot exciton trapping at the surface which causes non-radiative recombination. The authors should do a PLE experiment and measure the QY as a function of energy to verify if there are hot exciton surface effects^{19, 20}. Fig2c suggests that there is faster nonradiative decay by an excitation induced process.

Response:

We thank the reviewer for the comments on the sublinear pump fluence-dependent PL behavior in the unannealed PNC films. Following a thorough investigation, we find that the decline in PL efficiency upon the elevation of CW pump fluence does not stem from the intrinsic photophysics of PNCs, including multi-exciton Auger recombination and hot exciton effect. Conversely, the process is irreversible; that is, when the CW light fluence is reduced to the previous level following the entire in-situ PL experiment period, the PL intensity cannot fully restore (Figure 4a of the previous manuscript), which is attributable to photodegradation (*e.g.*, ligand detachment and QD aggregation/decomposition).

The issue of photodegradation brings about substantial difficulties in evaluating the impact of the operational conditions on charge recombination kinetics. In order to overcome this problem, ultrastable PNC films have been fabricated, which do not undergo irreversible alterations upon long-term CW light irradiation. The present work focuses on the PL behavior of stabilized PNC films, where the sublinear pump fluence dependence is absent (Figure 4b). In the revised manuscript, we have demonstrated that the multi-exciton effect is negligible under CW excitation, and the enhanced superlinear PL features can be attributed to the trap-state and charge-carrier interaction.

Revisions:

Page 6: *A series of representative steady-state PL spectra with varied CW pump fluences are presented in Figure 4a (see Figure S12 for complete data). The PL intensity increases with the pump fluence while the spectral profile remains consistent. A closer inspection uncovers a superlinear relationship between the PL intensity and the CW pump fluence, as illustrated in Figure 4b and its inset. For instance, by multiplying the PL signal measured at 0.64 mW/cm² with twenty, it is still significantly lower than that measured at 12.74 mW/cm². Such a superlinear dependence indicates that the PLQY is enhanced with the elevation of the excitation intensity.*

Page 8: *The landscape of operando charge recombination revealed by in-situ TRPL is illustrated in Figure 5f, where $n_{0,i}$ and δn denote the charge carriers generated by the CW beam and the pulse beam, satisfying the perturbation requirement, $\delta n \ll n_{0,i}$. The decay of δn involves two pathways: trap state-mediated nonradiative charge recombination (dashed arrow) and radiative charge recombination (solid arrow), which correspond to the first and second terms, respectively, in the continuity equation $d\delta n/dt = -k_1\delta n - 2k_2n_0\delta n$. With the*

elevation of the CW pump intensity, the value of n_0 is increased, which is anticipated to accelerate the TRPL decay; by contrast, the density of empty trap states (i.e., nonradiative recombination centers) is decreased owing to the trap filling by charge carriers, thus resulting in a reduced k_1 and a retarded recombination kinetics. Under operational conditions, the density of charge carriers is several orders of magnitude smaller than the trap state density³²⁻³⁵. This indicates that the effect of photoexcitation on filling trap states is dramatically greater than that on increasing n_0 . As a consequence, the in-situ spectroscopy results demonstrate a concomitant increase in the steady-state PL intensity and prolongation in the TRPL lifetime with the enhanced CW pump fluence, which mirrors the landscape of charge recombination in PNC films under realistic operational conditions that is not revealed by typical TRPL techniques.

8. Fig2. The in situ t=PL is different from the dispersion. Why is this? Mainly the in situ is single exponential whereas the dispersion is bi-exponential as is commonly seen. Why the single exponential?

Response:

We thank the reviewer for this pertinent enquiry. As the time scales for the typical and in-situ TRPL results are not aligned (0 – 170 ns vs. 0 – 36 ns, Figure 2c and Figure 2d in the previous manuscript), the latter appears to be more consistent with single exponential functions. As seen in Figure 5b of the revised manuscript, the in-situ TRPL kinetics demonstrate a high degree of alignment with the bi-exponential function. In addition, quantitative fitting results based on the bi-exponential functions are provided in Figure 5c and the relevant text.

Revisions:

Page 8: *A representative in-situ TRPL data in conjunction with the biexponential fitting results are depicted in Figure 5b, where the fast (τ_1) and slow (τ_2) decay components have been suggested to arise from charge recombination at the grain interface/surface and within the volume of grains, respectively³². As the CW pump intensity is elevated from 0.64 to 12.74 mW/cm² (Figure 5c), the value of τ_1 increases by a factor of 37% (1.85 ns \rightarrow 2.53 ns) while the variation in τ_2 is less than 13% (see complete fitting results in Figure S16 and Table S2). Concurrently, the weight of interface/surface recombination is reduced from 78% to 60%.*

9. Fig2c and d should be plotted on the same time window for better comparison

Response:

We thank the reviewer for this great suggestion. In the revised manuscript, both typical and in-situ TRPL data now share an identical time scale (Figure 4c, Figure 4d, Figure 5a, Figure 5b).

Revisions:

Page 6 & Page 7:

10. Fig2e shows a shape change to the spectrum. It looks like there is an ASE peak. Is there a peak developing? Perhaps a subtraction procedure would be useful.

Response:

We thank the reviewer for the insightful comments regarding the potential incorporation of ASE. As shown in **Figure R4**, the normalized PL profiles exhibit excellent overlap, excluding the occurrence of ASE. Meanwhile, under a CW pump intensity as low as $\sim 10^2$ mW/cm², it is fairly difficult to attain population inversion.

Figure R4. Normalized PL spectra with different CW pump intensities (corresponding to Figure 2e of the previous manuscript).

11. I am confused about the description of data in Fig2. The ex-situ data are in a film or a dispersion? Is it that in site and ex situ are for a film or for a device?

Response:

We thank the reviewer for highlighting the description issue. All the steady-state and time-resolved PL data in Figure 2 of the previous manuscript are collected on PNC films. In the revised manuscript, the tested sample information has been added in the figure caption for all spectroscopic data.

Revisions:

Page 5: Figure 3 (a) Schematic illustration of the integrated in-situ PL spectroscopy platform for the simultaneous acquisition of steady-state PL spectra and TRPL kinetics (see text for details). BS: beam splitter, DM: dichromatic mirror, LPF: long-pass filter, M: mirror, NDF: neutral density filter, PMT: photomultiplier tube. Representative experimental results of (b) in-situ PL spectra and (c) TRPL kinetics of the PNC film measured in a synchronous manner.

Page 6: Figure 4 (a) Steady-state PL spectra of the PNC film measured by in-situ PL spectroscopy at varied CW pump fluence. (b) Corresponding PL intensity versus CW pump fluence, which displays superlinear dependence. Inset: comparison of the PL spectra measured with excitation intensity values of 0.64 and 12.74 mW/cm², respectively, where the former spectrum has been multiplied by a factor of twenty. Pump fluence-dependent PL decay kinetics of the PNC film measured by (c) typical TRPL and (d) in-situ TRPL. The numbers denoted in panel (c) correspond to the fluence of the femtosecond pulse beam. In panel (d), the pump intensities of the CW beam for the presented TRPL data are 0.64, 1.27, 6.37, and 12.74 mW/cm², respectively. For the sake of comparison, the baselines of in-situ TRPL data have been subtracted, and the peak intensities have been normalized.

Page 7: Figure 5 (a) CW pump intensity-dependent in-situ TRPL kinetics for the colloidal PNC solution. (b) Representative in-situ TRPL kinetics (solid line) of the PNC film measured at a CW pump intensity of 0.64 mW/cm² and the corresponding fitting results (dashed line), where τ_1 and τ_2 are the lifetimes of the fast and slow components derived from the biexponential function. (c) Lifetime values of the fast (green) and slow (orange) decay components as a function of CW pump intensity. Also denoted in percentage are the weights of the slow component for each set of in-situ TRPL data. (d) Wide-field PL images of the PNC film measured with desired excitation power and integrating time. Scale bar: 5 μ m. Schematics for illustration of (e) the morphological evolution during PNC film fabrication and (f) the operando recombination mechanism with different CW excitation levels.

12. The t-PL kinetics should be fit to a biexponential and report of the two time constant and amplitudes.

Response:

We thank the reviewer for this constructive suggestion. The results of bi-exponential fitting, including both the time constants and amplitudes of the two exponential components, are presented in the revised manuscript (**Figure 5b**, **Figure 5c**, **Figure S16**, **Table S2**).

Revisions:

Page S17:

Supplementary Fig. 16. In-situ TRPL decay kinetics of the PNC film under different CW pump intensities fitted to the bi-exponential mode.

Supplementary Table 2. Fitting results of in-situ TRPL kinetics derived from the results shown in Supplementary Fig. 16.

Pump intensity	0.64 mW/cm ²	1.27 mW/cm ²	6.37 mW/cm ²	12.74 mW/cm ²
A_1	0.78	0.75	0.66	0.60
τ_1 (ns)	1.85	1.92	2.31	2.53
A_2	0.22	0.25	0.34	0.40
τ_2 (ns)	32.70	33.54	36.87	35.6.
$\langle \tau \rangle$ (ns)	8.70	9.80	13.91	15.65

13. They switch from a fs pulse to a ps pulse without much explanation. Why was this done? Where are the pulse durations and pump wavelengths? Both should be reported. They discuss the non-perturbative regime. That is not relevant. The main thing is to be in the single exciton regime where $\langle N \rangle < 1$. There are many ways to guarantee this situation for fs excitation^{14, 21-23}. But why switch to ps excitation. Moreover for pulsed excitation the unit should be fluence in $\mu\text{J} / \text{cm}^2$ not mW/cm^2 . It is a pulse with an energy not a power.

Response:

We thank the reviewer for the comments on the excitation source of the TRPL measurements.

For the typical TCSPC experiment, a fs pulsed laser (PHAROS, 190 fs, 6000 Hz) is employed in conjunction with an optical parametric amplifier (OPA, ORPHEUS) to generate the excitation beam at 450 nm. The pump intensity of the femtosecond laser can be viably tuned over three orders of magnitude, thereby enabling the acquisition of pump fluence-dependent TRPL kinetics (see **Figure 1c** for PNC solutions and **Figure 4c** for PNC films).

For the in-situ, operando TCSPC experiment, the excitation source was substituted by a ps laser diode (Edinburgh Instruments EPL-405, ~100 ps, 5 MHz). In this case, as long as the multi-exciton Auger recombination caused by the pulse laser is suppressed, its intensity tunability is not important because the carrier population state is determined by the CW light. However, the sample is co-excited by a strong CW light and a weak pulse light, as shown in **Figure 3c**. The CW light generates TRPL baseline and noise, while the pulse light generates the time-resolved signals; consequently, it requires a high repetition rate of the laser to shorten the accumulation time and simultaneously guarantee the signal-to-noise ratio.

The rationale for changing the low-frequency fs laser to the high-frequency ps laser has been delineated in the revised manuscript. In addition, the pump intensity unit of the pulse beams has been revised from mW/cm^2 to nJ/cm^2 or $\mu\text{J}/\text{cm}^2$, respectively, while the intensity of the CW beam remains in the of mW/cm^2 unit.

Revisions:

Page 6: Figure 4

Page 5: It is worth noting that the femtosecond laser has been replaced with a picosecond

laser diode used as the pulse beam source for in-situ TRPL testing. The picosecond laser diode possesses the capability to operate at a much higher frequency (5 MHz) than the femtosecond laser (6 kHz), a crucial factor in ensuring the acquisition of high-quality in-situ TRPL data without significantly prolonging the accumulation time.

Page 10: *For typical time-resolved PL measurement, a femtosecond pulsed laser (PHAROS, 190 fs, 6 kHz), in conjunction with an optical parametric amplifier (OPA, ORPHEUS) were utilized to generate the excitation beam at 450 nm.*

Page 10: *In detail, a continuous laser diode (Laserland, Wuhan RLTC Technology Co., Ltd., 1875A-450D-80mw-5v) and a picosecond laser diode (Edinburgh Instruments EPL-405, ~100 ps, 5 MHz) were used to generate the CW beam (wavelength: 450 nm) and pulse beam (wavelength: 405 nm), respectively, whose intensities were strictly modulated by the neutral density filters (Daheng Optics, GCO-074M) to meet the criteria of perturbation.*

14. Fig3a shows the PLQY vs fluence for CW pumping. This is a fascinating result. It would be useful to consider the QD in terms of competition kinetics between k_{rad} and k_{nonrad} . Why not plot the k_{rad} and k_{nonrad} as a function of temperature since you have QD and k_{decay} . Similarly Fig3b shows the lifetime vs temperature. It should be the rate constant not the lifetime. Because the rate constant is linear in scale and the time constant is nonlinear. So the rate constant a better measure. Examining the rate constants vs temperature in an appropriate plot can be very revealing as shown here for unraveling excitonic spatial coherence.

Response:

We thank the reviewer for the constructive suggestions concerning the pump fluence-dependent PLQY and the temperature-dependent TRPL experiments.

In the previous manuscript, the PLQY and temperature-dependent PL spectra were investigated with the aim of elucidating the disparate charge recombination mechanisms between the unannealed and annealed PNC films (*e.g.*, the variation in exciton binding energy). However, as replied to Comment-7, the following investigations have revealed that the majority of the PL decline caused by CW excitation is irreversible, as attributed to undesired photodegradation, rather than the anticipated multi-exciton Auger recombination. Therefore, the revised manuscript focuses on the improved PNC film with excellent stability against the CW and pulsed illuminations. The relevant content to which the reviewer refers has been removed from the current manuscript to maintain the content concise and the logic structure compact. The approach to studying excitonic spatial coherence by temperature-dependent TRPL kinetics (P. Kambhampati, *et al. Nano Lett.*, **2024**, *24*, 61), as mentioned by the reviewer, is both intriguing and useful, which will be showcased in our ongoing research work on the ultrafast excitonic dynamics in PNCs.

15. Fig3c-d show the Intensity vs temperure $1/T$. This is a useful plot but it could be done better. The bright colors are distracting and not helpful to reading the data. I imagine the purpose of the temperature dependence is to see if there is some activation barrier to going from emitting states to non emitting states. Such a plot is best done as an Arrhenius plot. The temperature dependence should also report on the energy shifting and bandwidth narrowing vs temperature to unravel details relevant to this situation.

Response:

We thank the reviewer for the comments regarding the temperature-dependent PL behavior. By fitting the temperature-dependent PL intensity to the Arrhenius equation, an activation energy can be obtained as mentioned by the reviewer. Furthermore, we concur with the reviewer's assertion that the variation in the emission peak and PL bandwidth against temperature is meaningful and can be applied to quantitatively analyze the fine excited-state structure and the phonon-electron interaction in PSCs. However, as outlined in the response to Comment-14, the temperature-dependent PL results have been removed in the reorganized manuscript.

16. The authors discuss the role of the surface and of surface ligands in controlling the PL kinetics. This point has been addressed in detail in CdSe QD. The idea is that there are surface excitons that are shallow so are not populated at 300K. By 30K these shallow states become populated and emit. Their emission is broadened and redshifted from the exciton emission due to phonon progressions. By virtue of being at the interface of the QD core and the ligands, the interface is very sensitive to ligands for the PL behavior.

Response:

We thank the reviewer for the enlightening insights into the temperature-controlled surface exciton population and emission in QDs. It is unambiguous that the low-temperature TRPL kinetics will prove to be of significant utility, as detailed in the recommended references (P. Kambhampati *et al.*, *Phys. Rev. B*, **2013**, *87*, 081201; *J. Chem. Phys.*, **2013**, *138*, 204705; *Chem. Phys.*, **2015**, *446*, 92), for estimating the contribution of surface excitons on the assembled PL feature. However, as stated in response to Comment-14 and Comment-15, the temperature-dependent PL data have been removed from the revised manuscript according to the reconstructed logic structure. For the sake of simplicity, we therefore would prefer not to delve into the temperature-dependent behavior in the current stage.

Response to the Reviewer-3's comments

This manuscript presents a novel and potentially impactful methodology for studying photoluminescence (PL) dynamics in perovskite quantum dot (PQD) films under operational conditions. The authors introduce an integrated “in-situ” spectroscopy setup that enables synchronous acquisition of steady-state and time-resolved PL (TRPL) under continuous-wave (CW) illumination, offering a more realistic picture of device-relevant light emission than conventional TRPL. This approach addresses a well-known limitation in the field, where ultrafast pulsed excitation used in TRPL experiments does not accurately reflect steady-state operational behavior. If validated and reproducible, this setup could become a valuable characterization tool for PQDs and other light-emitting materials.

1. The methodology is promising and generally sound. However, several points require clarification or additional evidence. First, the effect of CW intensity on the TRPL decay should be more rigorously assessed. The influence of laser pulse intensity is discussed, but a more detailed kinetic analysis would strengthen the conclusions. For example, the use of biexponential fitting alone may not fully capture the complexity of carrier dynamics, especially in systems where trap-assisted recombination or Auger processes may play a role. It would be beneficial to supplement the existing data with kinetic modeling to confirm that the derived time constants are consistent with the proposed mechanisms.

Response:

We thank the reviewer for the constructive suggestions on improving the rigorousness by kinetic modelling. In the revised manuscript, it is first experimentally verified that the charge recombination in annealed PNC films is distinct from that in highly dispersed PNC solutions (**Figure 5a** and **Figure 5b**). Subsequently, we present an in-depth quantitative analysis starting from the equilibrium-state classical continuity equation: $dn_0/dt = G - k_1n_0 - k_2n_0^2 = 0$, which incorporates the terms of both nonradiative trap-assisted recombination and radiative recombination. Multi-exciton Auger recombination, which may occur under the excitation of high-intensity pulse beams, is negligible for in-situ TRPL measurements, as derived from the pump fluence-dependent data (**Figure 4a-4d**). In the case of perturbation ($\delta n \ll n_0$, see evidence in **Figure S11**), the continuity equation turns out to be: $d\delta n/dt = -k_1\delta n - 2k_2n_0\delta n$, in line with the first-order reaction model (see detailed derivation on **Page 7**), resulting in an exponential profile for $I_{PL}(t)$. Furthermore, by taking the inhomogeneity factor into account, the TRPL kinetics are generally fitted with multi-exponential or stretched exponential functions. In our case, the experimental data can be adequately modelled by biexponential functions, thereby corroborating the recommendations proposed by Reviewer 2 (Comment 12). The two exponential components obtained are attributed to charge recombination at the grain interface/surface and within the volume of grains, respectively (J. Huang, *et al. Science*, **2015**, *347*, 519; N.-G. Park, *et al. Nat. Energy*, **2016**, *1*, 1). The relevant revisions per the reviewer's suggestion are summarized below.

Revisions:

Page 7: In quantitative, provided that multi-body Auger recombination is negligible, the equilibrium-state classical continuity equation in polycrystalline perovskites is typically expressed as: $dn_0/dt = G - k_1n_0 - k_2n_0^2 = 0$, where n_0 is the carrier density at the equilibrium state, t is time, G is the rate of photocarrier generation, k_1 is the rate constant for first-order nonradiative recombination, and k_2 is the rate constant for bimolecular recombination. Once a perturbation is superposed on the equilibrium state, the expression should be rewritten as: $d(n_0 + \delta n)/dt = G - k_1(n_0 + \delta n) - k_2(n_0 + \delta n)^2$, in which δn is the density of excess carrier with a value much smaller than equilibrium-state carrier density in a perturbation case ($\delta n \ll n_0$). A combination of the two aforementioned equations results in the classical continuity equation being reduced to the following form: $d\delta n/dt = -k_1\delta n - 2k_2n_0\delta n$, which is indicative of first-order reaction kinetics. In-situ TRPL experiment fulfills the prerequisites of perturbation, therefore the time-dependent PL intensity is predicted to be described by the first-order kinetics model with a decay lifetime of τ , $I_{PL}(t) = I_{PL}(0) \cdot \exp(-t/\tau)$. Furthermore, in consideration of the inhomogeneity of the chemical environment for individual perovskite grains, it has been proposed that the TRPL data be fitted more accurately with a bi-exponential function²⁴ or a stretched exponential function²⁵. Here we adopt the bi-exponential model for elucidation due to its higher fitting goodness than the stretched exponential model (Figure S15).

Page 7: A representative in-situ TRPL data in conjunction with the biexponential fitting results are depicted in Figure 5b, where the fast (τ_1) and slow (τ_2) decay components have been suggested to arise from charge recombination at the grain interface/surface and within the volume of grains, respectively^{26,27}.

2. A key point requiring clarification is the role of the CW light in modifying the sample during measurement. The irreversible changes observed after pre-illumination (Figure 4) suggest that CW exposure alters the material in a non-trivial way. This raises concerns about whether the in-situ TRPL truly reflects steady-state behavior, or instead captures a transient state of a material in flux. Can the authors demonstrate that measurements are reproducible on the same sample, and between different samples prepared under identical conditions? This would bolster the validity of the method and the conclusions drawn.

Response:

We thank the reviewer for the important comments on the stability and reproducibility of the spectroscopic studies. As stressed by the reviewer, the in-situ TRPL kinetics cannot accurately reflect the operando features, if irreversible changes take place during the measurements. Based on in-depth exploration, it is verified that the CW light intensity-dependent irreversible PL and TRPL variations arises from the chemical instability of PNC films used in the previous manuscript. Therefore, the tested samples have been redesigned to guarantee their operational stability, and a series of experiments have been conducted to verify the experiment reproducibility, as detailed below:

- (1) We propose a two-step post-synthesis treatment method, comprising annealing and preillumination, to prepare ultrastable PNC films. Thermal annealing is suggested to remove excess organic ligand, thus preventing the occurrence of irreversible PL decline related to photo-induced ligand detachment of the dispersed perovskite quantum dots (Figure 4a of the previous manuscript). Concurrently, the preillumination treatment can improve both the PLQY and stability of the sintered PNC films in terms of defect self-healing, thereby circumventing the irreversible PL enhancement of the annealed PNC film (Figure 4c of the previous manuscript).

Figure R5. PL intensity evolution of the PNC films with different treatments under continuous light illumination (corresponding to **Figure 2f**).

As verified by in-situ PL spectroscopy and summarized in **Figure R5**, the two-step treated film possesses excellent photostability for subsequent optical characterizations.

- (2) To guarantee that all the experimental findings are reversible and reproducible, we have carefully controlled the CW light intensity to be lower than 15 mW/cm^2 .

Figure R6. Steady-state PL spectra (left) and TRPL kinetics (right) of the PNC film before and after a complete experimental period (corresponding to **Figure S13** and **Figure S14**, respectively).

Both the steady-state PL spectra and TRPL kinetics for the same PNC film before and after a complete experimental period have been measured for quantitative comparison (**Figure R6**). The variations in PL intensity and TRPL lifetime are less than 5%, in great line with the robustness of the improved sample in spectroscopy measurements.

- (3) The reproducibility of the in-situ spectroscopy can be further verified by the stable real-time PL spectra (**Figure 3b**) and TRPL kinetics (**Figure 3c**).

Figure R7. Variation of the (a) PL spectra and (b) corresponding peak counts of the PNC film during in-situ PL and TRPL measurements (corresponding to **Figure S8**).

Figure R8. In-situ TRPL kinetics recorded during an entire experimental period with different CW illumination durations (corresponding to **Figure S9**).

In quantitative, the variations in PL intensity and decay lifetime (**Table S1**) are also less than 5%, which strongly manifest the reproducibility for both steady-state and time-resolved measurements.

Revisions:

Page 4: *Figure 2f displays the PL intensity evolution versus illumination time for the three types of PNC films, as derived from the long-term in-situ PL spectroscopy measurement (Figure S7), which unambiguously manifests the efficacy of the sequential annealing and preillumination treatment on optimizing the photostability.*

Page 5: *Figure 3b displays the as-obtained in-situ PL spectra, which are centered at 535 nm regardless of illumination period, aligning closely with the standard PL spectrum shown in Figure 2e. The fluctuation of PL intensity is found to be less than 2% over the entire experimental period (Figure S8), validating the prominent stability of the integrated PL spectroscopy platform.*

Page 5: *Concomitant with the in-situ PL spectra measurement, TRPL results are obtained by means of TCSPC. Three sets of TRPL kinetics with the same accumulation time of ~660 s, corresponding to the three regions denoted in in-situ PL spectra as exemplified in Figure 3c, share the identical time-resolved features.....Secondly, the three sets of TRPL kinetics are self-similar with a variation of average lifetime as small as ~4% (Figure S9 and Table S1), indicative of excellent reproducibility for the in-situ TRPL test.*

3. Further experimental details would also enhance reproducibility and interpretation. The authors should provide the repetition rate of the pulsed laser and explain how they confirmed that charge accumulation is avoided between pulses. Figure S8 is used to argue that the pulsed beam does not affect the observed kinetics; however, time constant values are not provided, and should be included to support this claim.

Response:

We thank the reviewer for the constructive suggestions. The repetition rates of the fs and ps lasers are 6 kHz and 5 MHz, respectively, which have been added to the Experimental Section of the revised manuscript. The interval between two pulses is therefore several orders of magnitude longer than the PL lifetime ($\sim 10^1$ ns), which is imperative for avoiding charge accumulation.

Figure R9. In-situ TRPL kinetics as a function of repetition rate for the ps laser.

Figure R9 illustrates the TRPL kinetics with different repetition rates. It is evident that the TRPL kinetics are not altered with the repetition rate, indicating the absence of charge accumulation.

In addition, per the reviewer's suggestion, the fitting results as well as the decay time constants obtained for the in-situ TRPL kinetics with different pulse intensities have been provided in the revised manuscript. The variation in the time constant is less than 3%, thereby verifying that the weak ps pulses do not introduce any pump fluence-dependent TRPL behavior (see below).

Revisions:

Page 5: *More importantly, the picosecond pulse intensity is sufficiently weak, so as not to pose interference to the steady-state PL spectra measurement (Figure S10) nor to introduce any pump fluence-dependent TRPL behavior (Figure S11).*

Page 10: For typical time-resolved PL measurement, a femtosecond pulsed laser (PHAROS, 190 fs, 6 kHz), in conjunction with an optical parametric amplifier (OPA, ORPHEUS) were utilized to generate the excitation beam at 450 nm.

Page 10: In detail, a continuous laser diode (Laserland, Wuhan RLTC Technology Co., Ltd., 1875A-450D-80mw-5v) and a picosecond laser diode (Edinburgh Instruments EPL-405, ~100 ps, 5 MHz) were used to generate the CW beam (wavelength: 450 nm) and pulse beam (wavelength: 405 nm), respectively, whose intensities were strictly modulated by the neutral density filters (Daheng Optics, GCO-074M) to meet the criteria of perturbation.

Page S12:

Supplementary Fig. 11. In-situ TRPL of PNC films as a function of pulsed excitation intensity, where the intensity of the CW beam remains unchanged (~ 7 mW/cm²). The circles are experimental data, and the solid curves are bi-exponential fitting results. The pulse intensities and as-obtained decay time constants have been displayed in the corresponding panels.

4. A few specific points are noted below:

Small changes: • Figure 2a and line 90: Figure says unannealed, while line 90 says annealed.
• Figure 4. b) and d) is practically impossible to distinguish between the solid line and the dotted line. Change colour scheme. • Line 130, accompanied..

Response:

We thank the reviewer for pointing out these errors and inaccuracies. Following a thorough reorganization and revision of the manuscript, all typographical and formatting errors have been rectified.

Revisions:

The *main text* have been thoroughly revised according to the reviewer's suggestions.

5. For Figure 2, the conditions (pump intensities) for CW and TRPL measurement are quite different, which implies different charge carrier generation and recombination rates (DOI: 10.1063/1.5143121) and can potentially explain differences in the trends presented in the paper. On the other hand, “in-situ TRPL” measurements employ CW illumination as a base, which they show can affect the PL response of the system over time (Figure 4). If the sample changes over time due to the CW, how can we rely on the “in-situ” TRPL information obtained? Can the authors show that the “in-situ” measurements are reproducible? Can the measurements be repeated in the same system already measured and obtain the same results? If not, can the experiments be repeated and obtain the same behaviour?

Response:

We thank the reviewer for these valuable inquiries, which have been addressed through the implementation of additional experiments in the revised manuscript, as detailed below.

- (1) As the reviewer has noted, the TRPL features are found to be strongly dependent on the pump intensity. The experimental condition of typical time-resolved studies (*i.e.*, under the excitation of ultrashort pulses) significantly differs from the operational conditions of PNC devices (*i.e.*, under the excitation of CW light). Therefore, the former may not precisely reflect the case in realistic PNC materials. In the revised manuscript, we performed two comparative studies: (i) the different behaviors between the typical and in-situ TRPL kinetics are attributed to the different photoexcitation conditions, in line with the reviewer’s consideration; (ii) the different behaviors between the dispersed PNC solution and the close-packed PNC film are attributed to the variation in the microscopic structure.
- (2) By improving the sample fabrication procedures, the issue of PL instability has been settled, as validated in **Figure 2f**, **Figure S13**, and **Figure S14**. In addition, the reproducibility of the in-situ spectroscopy platform has been solidly verified upon a series of confirmatory experiments in the revised manuscript (**Figure S8**, **Figure S9**, and **Table S1**). Detailed information can be found in the response to the second comment from Reviewer-3.

Revisions:

Page 2: ... *it is recognized that typical TRPL techniques are perhaps incapable of revealing the landscape of charge recombination in PNC devices under real operational conditions. The working devices are pumped by stable continuous-wave (CW) light or constant electric field, while the TRPL measurements are performed under the excitation of ultrafast pulses.*

Page 7: *As illustrated in Figure 5a, the in-situ TRPL kinetics of the PNC solution present negligible alteration in response to a wide range of CW pump intensity variations. No additional rapid decay component is observed at high pump fluence, in contrast to the typical TRPL data shown in Figure 1c, indicating that the multiexciton effect is minimized under CW photoexcitation. The distinctive in-situ TRPL behaviors facilitate the ascription of the retarded TRPL decay to the specific microscopic structure of PNC films, whose formation is*

accompanied by the transition from highly dispersed-state nanoparticles in solution (Figure 1a) to sintered close-packed large grains in solid-state configuration (Figure 2c). In the meantime, the improved PLQY at high CW pump fluence in PNC films is analogous to the findings reported in hundred-nanometer-sized polycrystalline perovskites²¹.

Page 5: Figure 3b displays the as-obtained in-situ PL spectra, which are centered at 535 nm regardless of illumination period, aligning closely with the standard PL spectrum shown in Figure 2e. The fluctuation of PL intensity is found to be less than 2% over the entire experimental period (Figure S8), validating the prominent stability of the integrated PL spectroscopy platform.

Page 5: Concomitant with the in-situ PL spectra measurement, TRPL results are obtained by means of TCSPC. Three sets of TRPL kinetics with the same accumulation time of ~660 s, corresponding to the three regions denoted in in-situ PL spectra as exemplified in Figure 3c, share the identical time-resolved features.....Secondly, the three sets of TRPL kinetics are self-similar with a variation of average lifetime as small as ~4% (Figure S9 and Table S1), indicative of excellent reproducibility for the in-situ TRPL test.

6. The authors claim the following (line 105): “According to what has been extensively documented in previous studies, the emergence of a rapid PL decay component at high pump fluence is attributed to the Auger recombination of multi-excitons, which, however, is ruled out due to the ultralow intensity of the pulsed laser in our case.” And use Figure S8 to show that the kinetics do not change by changing the intensity of the pulsed laser, but 2 questions emerge from this statement:
- (a) The authors employ high CW pump intensities. The power of illumination to excite a sample in a TRPL experiment has a strong influence on the concentration of photogenerated carriers, which in turn determines the type of recombination process that dominates in the system, including Auger recombination for high fluences ((DOI: 10.1039/d0c904950f), (DOI: 10.1063/1.5143121), (DOI: 10.1002/adfm.201910004). Have authors consider Auger recombination as a result of the CW illumination in the “in-situ” TRPL, or what was the criteria to discard this phenomenon as a result of the CW illumination?
 - (b) For Figure S8, the authors make the following statement: “The intensity of the pulsed beam has nothing to do with the observed in-situ TRPL kinetics in our case...”. However, it seems they do not provide time constant values for figure S8 that can corroborate this statement. I would recommend including this information.

Response:

We thank the reviewer for the important comments on the pump intensity-dependent TRPL behaviors.

- (1) The occurrence of Auger recombination under CW illumination is suggested to be minimized in the present work. Since Auger recombination is a multi-body process, if it plays a noneligible role under CW illumination in the present experiment, the TRPL kinetics should be accelerated by increasing the pump intensity, which is at odds with the experimental results (**Figure 4d**).
- (2) Per the reviewer’s suggestion, the fitting results of in-situ TRPL data with varied pulse intensities have been shown in **Figure S11** of the revised manuscript, displaying a fluctuation of decay time constant as small as ~3%.

Revisions:

Page 6: *Interestingly, the typical TRPL data summarized in Figure 4c elucidates accelerated decay kinetics at high pump fluence, which implies a more severe nonradiative exciton/charge recombination process, as is the case in PNC colloidal solution (Figure 1c). However, this is at odds with the increase in PLQY at high pump fluence as derived from steady-state PL spectra, which helps exclude the multi-exciton Auger recombination. Conversely, as visible in Figure 4d, the in-situ TRPL data demonstrate a more gradual decay profile when the intensity of the CW light is increased from 0.64 mW/cm² to 12.74 mW/cm². Notwithstanding the*

underlying mechanism that will be discussed later; in-situ TRPL kinetics are phenomenologically in accordance with the superlinear pump fluence-dependent PL spectra.

Page S12:

Supplementary Fig. 11. *In-situ TRPL of PNC films as a function of pulsed excitation intensity, where the intensity of the CW beam remains unchanged (~ 7 mW/cm²). The circles are experimental data, and the solid curves are bi-exponential fitting results. The pulse intensities and as-obtained decay time constants have been displayed in the corresponding panels.*

7. The authors are using a sum of exponentials to obtain 2 different time constants and then obtain an average value of the lifetime. What does this lifetime represent?

Response:

We thank the reviewer for the questions regarding the significance of the fitting results. As suggested by Reviewer 2 and in accordance with the reported studies (J. Huang, *et al. Science*, **2015**, 347, 519; N.-G. Park, *et al. Nat. Energy*, **2016**, 1, 1), the in-situ TRPL kinetics are fitted to bi-exponential functions, which yields two decay lifetimes related to charge recombination at the grain interface/surface and within the volume of grains, respectively.

Revisions:

Page 8: *A representative in-situ TRPL data in conjunction with the biexponential fitting results are depicted in Figure 5b, where the fast (τ_1) and slow (τ_2) decay components have been suggested to arise from charge recombination at the grain interface/surface and within the volume of grains, respectively³².*

8. It has been reported that the laser intensity influences the TRPL decay and the dominant recombination mechanisms, but this information is often hard to evaluate from a biexponential fitting (DOI: 10.1039/d0c904950f). Have the authors consider using a kinetic model to analyse the data and corroborate that the same conclusions can be reached?(DOI: 10.1002/adfm.201910004) (DOI: 10.1039/D0CP04950F)

Response:

We thank the reviewer for the constructive comments on the kinetic model for TRPL fitting. In the revised manuscript, having verified the negligible contribution of the multi-exciton Auger recombination, the quantitative analysis was initiated with the equilibrium-state classical continuity equation: $dn_0/dt = G - k_1n_0 - k_2n_0^2 = 0$, by taking the nonradiative trap-assisted recombination and radiative recombination into account (**Figure 4a-4d**). The in-situ TRPL operates in a perturbation case ($\delta n \ll n_0$, see evidence in **Figure S11**), leading to the transformation of the continuity equation to the first-order kinetic form, $d\delta n/dt = -k_1\delta n - 2k_2n_0\delta n$ (see detailed derivation on **Page 7**). It can thus be concluded that the integration equation is, in fact, an exponential decay function. In addition, the complexity arising from the film inhomogeneity may result in a derivation from the rigorous single exponential prediction of the TRPL decay profile. According to the proposal of Reviewer 2, which is also consistent with the fitting results, we select the bi-exponential function to describe the TRPL kinetics. From a statistical perspective, the as-obtained two exponential components are pertinent to charge recombination at the grain interface/surface and within the volume of grains, respectively (J. Huang, *et al. Science*, **2015**, 347, 519; N.-G. Park, *et al. Nat. Energy*, **2016**, 1, 1).

Revisions:

Page 7: *In quantitative, provided that multi-body Auger recombination is negligible, the equilibrium-state classical continuity equation in polycrystalline perovskites is typically expressed as: $dn_0/dt = G - k_1n_0 - k_2n_0^2 = 0$, where n_0 is the carrier density at the equilibrium state, t is time, G is the rate of photocarrier generation, k_1 is the rate constant for first-order nonradiative recombination, and k_2 is the rate constant for bimolecular recombination. Once a perturbation is superposed on the equilibrium state, the expression should be rewritten as: $d(n_0 + \delta n)/dt = G - k_1(n_0 + \delta n) - k_2(n_0 + \delta n)^2$, in which δn is the density of excess carrier with a value much smaller than equilibrium-state carrier density in a perturbation case ($\delta n \ll n_0$). A combination of the two aforementioned equations results in the classical continuity equation being reduced to the following form: $d\delta n/dt = -k_1\delta n - 2k_2n_0\delta n$, which is indicative of first-order reaction kinetics. In-situ TRPL experiment fulfills the prerequisites of perturbation, therefore the time-dependent PL intensity is predicted to be described by the first-order kinetics model with a decay lifetime of τ , $I_{PL}(t) = I_{PL}(0) \cdot \exp(-t/\tau)$. Furthermore, in consideration of the inhomogeneity of the chemical environment for individual perovskite grains, it has been proposed that the TRPL data be fitted more accurately with a bi-exponential function²⁴ or a stretched exponential function²⁵. Here we adopt the bi-exponential model for elucidation due to its higher fitting goodness than the stretched exponential model (Figure S15).*

Page 7: *A representative in-situ TRPL data in conjunction with the biexponential fitting results are depicted in Figure 5b, where the fast (τ_1) and slow (τ_2) decay components have been suggested to arise from charge recombination at the grain interface/surface and within the volume of grains, respectively^{26,27}.*

9. Could the authors provide the repetition rate employed for the laser? How did the authors corroborate that the repetition rate was adequate for the system, particularly at preventing charge accumulation in the material? (DOI: 10.1039/d0c904950f)

Response:

We thank the reviewer for drawing attention to the potential issues that may be caused by the high repetition rate in TRPL experiments. The repetition rates are 5 MHz and 6 kHz, respectively, for the ps and fs pulses, as provided in the revised manuscript.

Figure R10. In-situ TRPL kinetics as a function of repetition rate for the ps laser.

Furthermore, in order to ensure the suppression of charge accumulation, the TRPL kinetics are measured as a function of repetition rate for comparison. As shown in **Figure R10**, the TRPL kinetics are not influenced by the repetition rate, excluding the occurrence of charge accumulation in our experiments.

Revisions:

Page 10: For typical time-resolved PL measurement, a femtosecond pulsed laser (PHAROS, 190 fs, 6 kHz), in conjunction with an optical parametric amplifier (OPA, ORPHEUS) were utilized to generate the excitation beam at 450 nm.

Page 10: In detail, a continuous laser diode (Laserland, Wuhan RLTC Technology Co., Ltd., 1875A-450D-80mw-5v) and a picosecond laser diode (Edinburgh Instruments EPL-405, ~100 ps, 5 MHz) were used to generate the CW beam (wavelength: 450 nm) and pulse beam (wavelength: 405 nm), respectively, whose intensities were strictly modulated by the neutral density filters (Daheng Optics, GCO-074M) to meet the criteria of perturbation.

10. As mentioned in Figure 4, the samples seem to change under CW conditions over time, which could affect the results of “in-situ” TRPL. Could the authors comment on the influence of the CW on “in-situ” TRPL regarding sample relaxation/charge accumulation between laser pulses?

Response:

We thank the reviewer for the constructive comments.

First, in the previous manuscript, the observed sample variation during the long-term CW experiment has been verified to be photodegradation in our subsequent experiments. This complicates the evaluation of the impact of operational conditions on the intrinsic photophysics of PNCs. Therefore, as stated in the response to the second comment of Reviewer-3, we have improved the film fabrication and ensure that all the operando behaviors reported in the revised manuscript are irreversible and reproducible. This has been collectively supported by the photostability and spectral stability experiments (**Figure 2f**, **Figure 3c**, **Figure S8**, **Figure S9**, **Figure S13**, **Figure S14**).

Second, we concur with the reviewer’s assertion that the CW intensity-dependent TRPL behavior is attributable to the varied charge population. According to the derived kinetic model (see schematic illustration in **Figure 5f**), the enhanced charge population in trap states simultaneously gives rise to an increase in PL efficiency and a prolongation in TRPL decay time constant. However, we would like to stress that this is substantially different from the “charge accumulation issue” discussed in Comment 9. The charge accumulation caused by a high laser repetition rate has been excluded by **Figure R10**. In addition, if the CW light poses the issue of charge accumulation, the baseline of the in-situ TRPL kinetics (*i.e.*, TRPL signals before $t = 0$) should not be a constant, which is not reflected in the presented data (**Figure 3c**).

Response to the reviewers' comments:

Response to the Reviewer-1's comments

1. The revised version of the manuscript addresses many of the issues raised in the initial round of review, both by myself and by the other reviewers. The data are now presented more consistently, and the discussion is better structured. However, I must say that the revised paper feels like an entirely different manuscript. A major part of the original work — the time evolution of the device — has now been removed. As a result, the focus and scope have shifted significantly.

Response:

We are grateful to the reviewer for concluding that the manuscript has been improved by our revisions in terms of data presentation and discussion.

2. I still find the distinction between what the authors call "operando" and "non-operando" data rather unclear. In Figures 4 and 5, the two datasets are now presented on identical scales, which is good. However, they are background-corrected differently, and the operando data clearly contain dark counts not present in the non-operando measurements. This background difference — particularly the dark counts around negative times — appears to be the main source of the observed changes in decay behavior.

Response:

We thank the reviewer for reminding us of paying attention to the potential mistakes outcome in data analysis due to the different backgrounds in typical (*i.e.*, “non-operando”) and in-situ (*i.e.*, “operando”) TRPL data. As outlined in the Experimental Section of the manuscript, the essential distinction between the typical and in-situ TRPL measurements lies in the implementation of the CW actinic light superimposed on the pulsed measuring light.

Figure R1. Representative typical and in-situ TRPL traces displayed in a linear scale. (The data were taken from those illustrated in Figure 4b and Figure 4c of the manuscript.) (a) Original TRPL traces. (b) The pair of traces shown in panel (a) after shifting the baseline to zero. (c) Comparison of the background signals under the same ordinate.

In an ideal case devoid of any noise, the background signal at negative time ($t < 0$) is anticipated to be zero in the case of typical TRPL, or to be a positive constant owing to the biased CW light in the case of in-situ TRPL. Here, we note that the presence of a background signal in an in-situ TRPL trace conforms to the principle of in-situ measurement, reflecting the operational condition.

In reality, both typical and in-situ TRPL traces bear noise and dark counts as shown in Figure R1 (a), (b). It is seen unambiguously that the peak count of the typical TRPL trace is substantially higher than that of the in-situ TRPL trace, since the intensity of the pulsed laser has been strictly controlled in the latter to satisfy the “perturbation” criterion.

Figure R1 (c) compares the ‘noise and dark counts’ of the different kinds of TRPL traces under the same absolute ordinate. Clearly, the levels of ‘noise and dark counts’ are comparable between the pair of TRPL traces. **Therefore, either the background difference or the dark counts cannot inflict any aberration on the observed changes in decay behavior.**

Figure R2. TRPL traces shown in a linear scale. (a) Normalized traces. (b) Zoom in section of panel (a). Note the different ordinate scales of the pair of traces.

In the manuscript, we have normalized the TRPL traces and presented them on identical scales, so as to better illustrate the pump-intensity dependence of the TRPL kinetics. Consequently, with reference to the typical TRPL traces, both the background and the signal of in-situ TRPL traces are seemingly amplified. **Nevertheless, based on the above analysis and for the sake of presentation clarity, we would like to maintain the previous data presentation as employed in the manuscript.**

3. Overall, the differences between operational and non-operational conditions have become quite small. Yet, due to inconsistent background levels across datasets and a lack of rigorous analysis of this background, it remains extremely difficult to quantify these differences in any meaningful way.

Response:

The distinction between the types of TRPL data is embodied by the significantly different dependences of the PL kinetics on excitation intensities.

Figure R2. Excitation intensity-dependent PL decay kinetics measured by (a) typical TRPL and (b) in-situ TRPL techniques, respectively (corresponding to Figure 4c and Figure 4d in the main text).

As shown in Figure R2, with the enhancement of excitation intensity, **the decay kinetics are accelerated for the typical TRPL while retarded for the in-situ TRPL.**

The background issue has been comprehensively discussed in the above response, which manifests that the visual difference in the background level of the two types of TRPL measurements stems from the data processing method adopted, while the absolute background levels are actually comparable. Furthermore, given that the maximum TRPL counts are at least two orders of magnitude higher than the background counts, the impact of the background on the dependence of TRPL kinetics on excitation intensity is negligible.

4. Fundamentally, I struggle to see what new insight or capability the technique actually provides. The results are not convincingly quantified, and in the current form I cannot recommend the manuscript for publication. Even if the authors were to conduct a more robust background analysis and present more convincing data, I am not convinced the changes would be significant enough to yield meaningful conclusions about the material's behavior — certainly not at a level suitable for Nature Communications.

Response:

We thank the reviewer for pointing out the potential issue that the scientific insights seem unclear in the current manuscript. The significance of the present study can be demonstrated from the following two aspects:

- (1) **Operando charge recombination mechanism in PNC films.** A correct understanding of the charge recombination mechanism is crucial for the rational design and optimization of high-efficiency PNC materials. According to the widely reported studies, where recombination kinetics are characterized by traditional TRPL techniques, PL efficiency is suggested to decline upon the elevation of excitation intensity due to multi-exciton Auger recombination. This is generally accompanied by a shortened PL lifetime, consistent with the typical TRPL results presented in this manuscript. However, as demonstrated by the excitation intensity-dependent PL spectra, the PLQY increases with illumination. **Such a contradiction stems from the inconsistent testing conditions between the PL spectra and typical TRPL measurement:** the sample is excited by a CW beam in the former and by a high-energy pulsed laser in the latter.

To overcome this technical limitation, we propose the in-situ TRPL method, which allows for measuring the recombination kinetics under CW illumination in a perturbation case. Using this new technique, the TRPL lifetime is observed to increase with CW excitation intensity. This finding not only **rationalizes the CW light-induced PL efficiency enhancement** (see analysis and discussion in the manuscript) but also **essentially resolves the longstanding mismatch between typical pulsed TRPL results and the CW operational behavior.**

- (2) **Methodology of operando dynamics/kinetics characterization.**

Devolping operando characterization techniques, aiming to elucidating the mechanism of photochemical and photophysical reactions under real operational conditions, has become a cutting-edge challenge in the fields of photochemistry (*Nat. Commun.*, **2016**, 7, 11918), electrochemistry (*Nat. Energy*, **2018**, 3, 46), biochemistry (*Nat. Commun.*, **2022**, 13, 547), and photoelectronics (*Nat. Commun.*, **2023**, 14, 8000). For instance, regarding materials employed for light emission, by moving research from characterizing materials/molecules in a passive state to diagnosing active systems, it allows the accurate mechanisms affecting quantum efficiency and working stability in real-world applications to be precisely identified.

Furthermore, the proposed in-situ TRPL technique possesses superior sensitivity and

simplicity in comparison to other classical time-resolved techniques (*e.g.*, femtosecond transient absorption spectroscopy). These advantages, combined with the excellent stability and reproducibility (see verification in Figure 3b, Figure 3c, Figure S8, and Figure S9), are thus anticipated to provide the technique a variety of applications in relevant research areas.

Response to the Reviewer-2's comments

The authors have well revised the manuscript and answered the questions. This paper is suitable for publication with a minor addition.

Since the authors are discussing TRPL with a streak camera on LHP QD< I encourage them to cite the most recent and precise work on the subject published by Lopw

Response and revision:

We are grateful to the reviewer for the positive view of our work and for concluding that our work is ready for publication.

We also thank the reviewer for recommending a series of noteworthy references concerning the TRPL studies in PNC by means of the streak camera. In the revised manuscript, two of the recommended papers have been cited in the appropriate sections as listed below:

Page 2: *Time-resolved PL (TRPL) techniques, including streak camera^{9,10},...such as high temporal resolution, tunable detection window, and superior sensitivity^{15,16},...*

References:

[10] Strandell, D., Mora Perez, C., Wu, Y., Prezhdo, O. V. & Kambhampati, P. Excitonic quantum coherence in light emission from CsPbBr₃ metal-halide perovskite nanocrystals. Nano Lett. 24, 61–66 (2024).

[16] Strandell, D. P. & Kambhampati, P. Light emission from CsPbBr₃ metal halide perovskite nanocrystals arises from dual emitting states with distinct lattice couplings. Nano Lett. 23, 11330–11336 (2023).

Response to the Reviewer-3's comments

I am satisfied that the authors have fully addressed the concerns raised in the initial review. I believe the authors have addressed the concerns of the reviewers generally very well. The revisions provide clear experimental details, improved data presentation, and quantitative analysis that support the conclusions. The rationale for the “operando” CW-bias TRPL approach is better articulated - I believe it is potentially useful for the authors to consider adding a one-sentence definition early in the Introduction clarifying the use of operando to mean under steady-state CW optical conditions, emulating device illumination conditions, weather than any electrical bias.

The manuscript is consistent and much improved from the original manuscript - I believe this will be of benefit to the scientific field and nor warrants publication subject to perhaps some minor editorial corrections for clarity and consistency.

Response and revision:

We are grateful to the reviewer for the positive view of the revisions we made and for concluding that our work is ready for publication.

We also thank the reviewer for the suggestion of improving the clarity of operando measurements. Accordingly, the manuscript has been revised as below:

Page 2: *For example, color-conversion PNC devices are pumped by a stable continuous-wave (CW) light, while the TRPL measurements are performed under the excitation of ultrafast pulses.*

Page 2: *The sophisticated design of the photoexcitation method enables precise examination of the operando exciton/charge recombination kinetics of PNCs under the sustained excitation of a CW beam.*

Response to the reviewers' comments:

Response to the Reviewer-1's comments

1. I think the manuscript improved overall and the presentation is better. I can follow things now, and I find that the main point — that the trend is superlinearity under CW light and sublinearity under pulsed excitation — is clear. I think the authors convinced me on that point, and I agree this is probably a reflection of trapping, and that this approach can be used to study trapping. In a very stretched way, it can also be related to “operando” conditions, though that's limited to optical illumination, not a real device bias... I personally would not call it operando and most device people as well would be confused.

Response:

We sincerely thank the reviewer for acknowledging that the manuscript has been improved in terms of data presentation, particularly in highlighting the distinct TRPL behaviors observed under CW and pulsed excitation.

Regarding the definition of *operando* conditions, it varies depending on the type of device. For instance, in perovskite photovoltaics, the *operando* condition typically involves illumination under simulated sunlight with a specific load, whereas for perovskite LEDs, it refers to operation under a sustained DC voltage. These examples likely align with what the reviewer had in mind for the definition of “*operando*” condition, which typically incorporates an electric field (*e.g.*, the sunlight induced internal field in photovoltaics and the bias field applied by DC voltage in LEDs), which cannot be replicated in a purely optical experiment.

Besides, an important application of PNCs lies in color conversion device, where a PNC film is excited by CW short-wavelength light to generate PL at desired wavelengths. The proposed in-situ TRPL platform is specifically designed to characterize recombination kinetics under conditions that closely mimic the *operando* environment of such PNC-based color conversion devices.

This point was also raised by Reviewer 3 in the previous review round, who suggested: “*The rationale for the “operando” CW-bias TRPL approach is better articulated - I believe it is potentially useful for the authors to consider adding a one-sentence definition early in the Introduction clarifying the use of operandi to mean under steady-state CW optical conditions, emulating device illumination conditions, weather than any electrical bias.*” In response, we have revised the manuscript with a focus on color conversion applications. The relevant text now appears on Page 2 and reads:

“For example, color-conversion PNC devices are pumped by a stable continuous-wave (CW) light, while the TRPL measurements are performed under the excitation of ultrafast pulses. It is uncertain whether the observed phenomenon in TRPL experiments can accurately mirror the stable operational condition.”

2. The part about lifetime change is still shaky. The graph itself doesn't convince me. The authors really try to sell it, but the experiments are so different — illumination power, laser system, repetition rate, everything. You can't directly compare absolute lifetimes between these two regimes. It's okay to show this data, but it shouldn't be pushed as a key point.

Response:

We thank the reviewer for the comments on the different experimental parameters between the typical TRPL and the proposed in-situ TRPL measurements.

Indeed, the discrepancy in photoexcitation conditions between pulsed laser-based TRPL and CW-driven color conversion applications serves as the key motivation for developing the operando kinetics characterization method in this study. As the reviewer rightly pointed out, the carrier behavior under CW excitation cannot be adequately captured or interpreted using conventional TRPL; conversely, it aligns well with the results obtained through our in-situ TRPL approach.

A detailed discussion on this point can be found on Page 6 of the manuscript, which further supports the effectiveness of the in-situ TRPL technique in revealing recombination kinetics under operando-like conditions.